# Structural basis for transcription complex disruption by the Mfd translocase

Jin Young Kang[1†‡], Eliza Llewellyn[1†], James Chen[1], Paul Dominic B Olinares[2], Joshua Brewer[1], Brian T Chait[2], Elizabeth A Campbell[1], Seth A Darst[1]*

[1]Laboratory of Molecular Biophysics, The Rockefeller University, New York, United States; [2]Laboratory of Mass Spectrometry and Gaseous Ion Chemistry, The Rockefeller University, New York, United States

**Abstract** Transcription-coupled repair (TCR) is a sub-pathway of nucleotide excision repair (NER) that preferentially removes lesions from the template-strand (t-strand) that stall RNA polymerase (RNAP) elongation complexes (ECs). Mfd mediates TCR in bacteria by removing the stalled RNAP concealing the lesion and recruiting Uvr(A)BC. We used cryo-electron microscopy to visualize Mfd engaging with a stalled EC and attempting to dislodge the RNAP. We visualized seven distinct Mfd-EC complexes in both ATP and ADP-bound states. The structures explain how Mfd is remodeled from its repressed conformation, how the UvrA-interacting surface of Mfd is hidden during most of the remodeling process to prevent premature engagement with the NER pathway, how Mfd alters the RNAP conformation to facilitate disassembly, and how Mfd forms a processive translocation complex after dislodging the RNAP. Our results reveal an elaborate mechanism for how Mfd kinetically discriminates paused from stalled ECs and disassembles stalled ECs to initiate TCR.

**\*For correspondence:** darst@rockefeller.edu

†These authors contributed equally to this work

**Present address:** ‡Department of Chemistry, KAIST, Daejeon, Republic of Korea

**Competing interests:** The authors declare that no competing interests exist.

## Introduction

DNA template strand (t-strand) lesions that block elongation by RNA polymerase (RNAP), most notably UV-induced cyclobutane dimers (*Bohr et al., 1985*; *Witkin, 1966*), are targeted for preferential repair in a process called transcription-coupled repair [TCR; (*Hanawalt and Spivak, 2008*; *Mellon and Hanawalt, 1989*; *Mellon et al., 1987*)]. In TCR, the stalled elongating RNAP serves as an efficient detector of t-strand lesions that then becomes a privileged entry point into the nucleotide excision repair (NER) pathway through the action of a transcription repair coupling factor (TRCF). While putative TRCFs have been identified in archaea and eukaryotes (*Troelstra et al., 1992*; *Walker et al., 2017*; *Xu et al., 2017*), the bacterial superfamily 2 (SF2) Mfd translocase (*Smith et al., 2007*) has been shown in vitro and in vivo to be a bacterial TRCF (*Kunala and Brash, 1992*; *Oller et al., 1992*; *Selby and Sancar, 1991*; *Selby and Sancar, 1993*; *Selby et al., 1991*).

*Escherichia coli* (*Eco*) Mfd (mutation frequency decline – named for its activity to reduce the frequency of suppressor mutations that occur when protein synthesis is inhibited subsequent to mutagenic treatment) was discovered through genetic analysis (*Bockrath et al., 1987*; *Witkin, 1966*), then identified as a TRCF, purified, and biochemically characterized (*Selby and Sancar, 1993*; *Selby and Sancar, 1994*; *Selby and Sancar, 1995a*; *Selby and Sancar, 1995b*). These experiments showed that Mfd was necessary and sufficient for TCR in vivo and in vitro and that it expressed two major activities:

1. Relief of inhibition of NER by recognition and ATP-dependent removal of the stalled RNAP elongation complex (EC) otherwise concealing the damaged DNA (*Brueckner et al., 2007*; *Selby and Sancar, 1990*).

2. After RNAP displacement, stimulation of the rate of DNA repair by direct recruitment of the Uvr(A)BC endonuclease to the lesion (*Deaconescu et al., 2012*; *Selby and Sancar, 1990*; *Selby and Sancar, 1991*; *Selby and Sancar, 1993*).

Subsequent biochemical, biophysical, and structural analyses have painted a more detailed outline of the Mfd functional cycle as a TRCF:

1. Mfd engages with stalled or paused ECs (the specific means of EC stalling is not important) through two interactions: (i) Mfd binds to the RNAP β subunit via its RNAP-Interacting Domain RID; (*Deaconescu et al., 2006*; *Park et al., 2002*; *Selby and Sancar, 1995b*; *Smith and Savery, 2005*; *Westblade et al., 2010*), and (ii) Mfd binds to duplex DNA upstream of the EC in an ATP-dependent manner via its RecG-like SF2 translocase domains (*Chambers et al., 2003*; *Deaconescu et al., 2006*; *Gorbalenya et al., 1989*; *Mahdi et al., 2003*; *Selby and Sancar, 1995b*).
2. The Mfd SF2 ATPase translocates on the upstream duplex DNA, pushing the RNAP in the downstream direction (same direction as transcription) via multiple cycles of ATP hydrolysis (*Howan et al., 2012*; *Park et al., 2002*). Backtracked and arrested ECs (*Komissarova and Kashlev, 1997a*; *Komissarova and Kashlev, 1997b*) can thus be rescued into productive elongation (*Park et al., 2002*). However, if a t-strand lesion or other type of roadblock prevents RNAP forward translocation, the continued translocase activity of Mfd overwinds the upstream region of the transcription bubble, facilitating displacement of the RNA transcript, transcription bubble reannealing, and removal of the RNAP from the DNA (*Chambers et al., 2003*; *Deaconescu et al., 2012*; *Howan et al., 2012*; *Manelyte et al., 2010*; *Murphy et al., 2009*; *Park and Roberts, 2006*; *Park et al., 2002*; *Proshkin and Mironov, 2016*; *Selby and Sancar, 1995a*; *Smith and Savery, 2005*; *Smith et al., 2007*).
3. After disruption of the EC, Mfd remains on the DNA and continues to slowly translocate in the downstream direction in a highly processive manner over thousands of base pairs, carrying the RNAP along for the ride (*Graves et al., 2015*; *Haines et al., 2014*; *Howan et al., 2012*).
4. The processively translocating Mfd-RNAP complex disassembles when it interacts with UvrA$_2$B, leaving behind the UvrA$_2$B complex to recruit UvrC and complete the NER pathway (*Fan et al., 2016*; *Selby, 2017*).

Most of the biochemical activities of Mfd, such as ATPase activity, DNA binding and translocation, and interaction with UvrA, are strongly repressed in free Mfd (apo-Mfd) (*Manelyte et al., 2010*; *Murphy et al., 2009*; *Smith et al., 2007*). The apo-Mfd X-ray crystal structure revealed a protein with six structural modules connected by flexible linkers but held in a compact, inactive conformation (*Deaconescu et al., 2006*). Mfd activities are 'unleashed' by engagement with the EC, which is expected to be accompanied by profound conformational changes in Mfd (*Srivastava and Darst, 2011*).

Here, we used single particle cryo-electron microscopy (cryo-EM) to visualize how Mfd engages and displaces a stalled EC. We observe Mfd undergoing its ATP hydrolysis cycle attempting to release the RNA transcript and dislodge the RNAP from the DNA template. The ECs were stalled by nucleotide deprivation on a DNA scaffold containing a non-complementary transcription bubble that cannot rewind. Thus, despite engaging in cycles of ATP hydrolysis, Mfd was unable to efficiently dislodge the RNAP from the nucleic acids, facilitating the visualization of intermediates.

Using image classification approaches, we visualized seven distinct Mfd-EC complexes, some with ATP and others with ADP. Features of the structures allow their placement in a pathway that provides a structural basis for understanding the extensive remodeling of Mfd upon its engagement with the EC and displacement of the RNAP. The structures explain how Mfd is remodeled from the repressed conformation (*Deaconescu et al., 2006*), how the UvrA-interacting surface of Mfd is hidden during most of the remodeling process to prevent premature engagement with the NER pathway, how Mfd alters the RNAP conformation to facilitate disassembly, and how Mfd ultimately forms the processive translocation complex after dislodging the RNAP (*Graves et al., 2015*). Our results reveal an elaborate mechanism for how Mfd kinetically discriminates paused from stalled ECs and disassembles stalled ECs to initiate TCR, and provide insights into the molecular motions that initiate TCR.

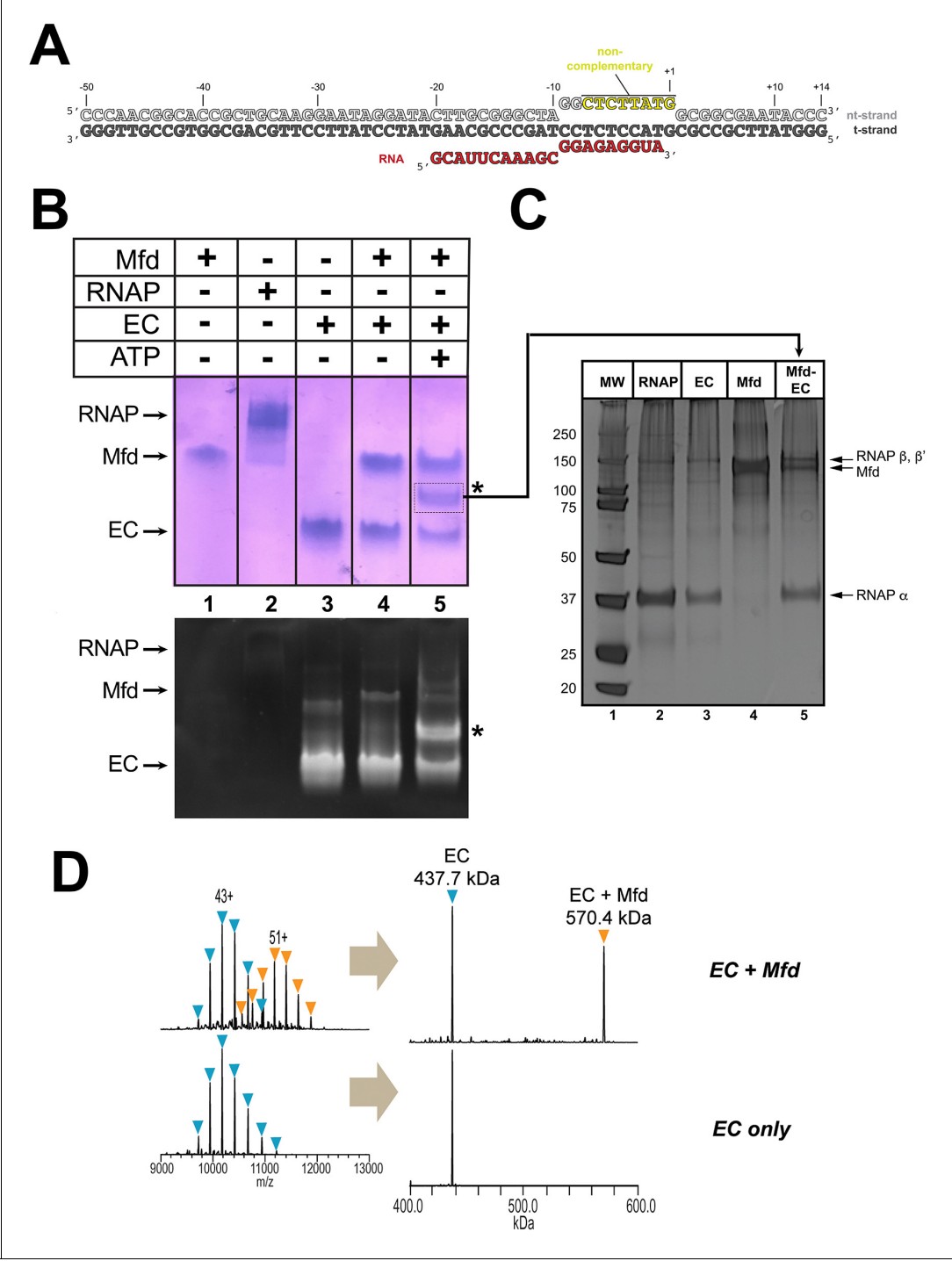

**Figure 1.** Formation of stable complexes between Mfd and an RNA polymerase (RNAP) elongation complex (EC). (**A**) The nucleic acid scaffold used for native mass spectrometry (nMS) and cryo-EM. The region of the nt-strand DNA shown in yellow is non-complementary with the t-strand. (**B**) Native gel electrophoretic mobility shift assay reveals the formation of a unique complex in the presence of an *Eco* RNAP EC (formed on the nucleic acid scaffold shown in **A**), Mfd, and 2 mM ATP (band labeled '*', lane 5). The same gel was stained with Coomassie blue to reveal protein (top panel) or Gel Red to reveal nucleic acids (bottom panel). (**C**) The band labeled '*' (panel B, lane 5) was excised from the gel and analyzed by SDS-polyacrylamide gel electrophoresis with silver staining, revealing the presence of RNAP and Mfd (lane 5). (**D**) nMS spectra and the corresponding deconvolved spectra for EC and Mfd-EC samples with the nucleic acid scaffold shown in (**A**). RNAP and the scaffold form a stable EC with 1:1 stoichiometry (437.7 kDa complex). Upon incubation of this complex with sub-stoichiometric Mfd (Mfd:EC ratio of 1:2) and 2 mM ATP, about 40% of the EC was converted to the Mfd-EC complex (570.4 kDa) with 1:1 stoichiometry. Excess EC was mixed with Mfd because unbound Mfd yielded extremely high relative peak signals that dominated the spectrum, making interpretation and quantitation difficult.

## Results

### Mfd with ATP forms a stable complex with an RNAP EC assembled on a non-complementary transcription bubble

We assembled *Eco* RNAP ECs on a nucleic acid scaffold (*Komissarova et al., 2003*) with the following features: (i) a 20-mer RNA transcript designed to generate a post-translocated nine base pair RNA/DNA hybrid with the DNA t-strand, (ii) a mostly non-complementary transcription bubble to inhibit Mfd-mediated displacement of the RNAP (*Park and Roberts, 2006*), and (iii) 40 base pairs of duplex DNA upstream of the transcription bubble to allow Mfd function (*Park et al., 2002*; *Figure 1A*). Native electrophoretic mobility shift analysis (EMSA) showed a unique band containing protein and nucleic acid that appeared only in the presence of the pre-formed EC, Mfd, and ATP (labeled '*' in *Figure 1B*). Subsequent analysis of the contents of the *-band by SDS-polyacrylamide gel electrophoresis showed that it contained both RNAP and Mfd (*Figure 1C*, lane 5). Stable ECs and Mfd-ECs with 1:1 stoichiometry were also detected by native mass spectrometry (nMS; *Figure 1D*).

### Seven structures in the Mfd activity cycle

In the crystal structure of the 130 kDa *Eco* apo-Mfd (*Deaconescu et al., 2006*), domains D1a-D2-D1b form a structural unit with similarity to the namesake elements of UvrB that interact with UvrA (the 'UvrB homology module'; *Figure 2A*). Indeed, mutagenesis and structural studies establish that UvrB and Mfd share a common mode of UvrA recognition (*Deaconescu et al., 2006*; *Deaconescu et al., 2012*; *Manelyte et al., 2010*; *Pakotiprapha et al., 2009*; *Pakotiprapha et al., 2012*; *Selby and Sancar, 1995b*). D3 is a non-conserved, lineage-specific domain with unknown function. D4 (RNAP interacting domain, or RID) is necessary and sufficient for Mfd interaction with the RNAP βprotrusion (*Deaconescu et al., 2006*; *Park et al., 2002*; *Selby and Sancar, 1995b*; *Smith and Savery, 2005*; *Westblade et al., 2010*). D5 and D6 comprise the RecG-like SF2 translocase domains (TD1/TD2). The interaction between the C-terminal D7 and the UvrA-interacting surface of D2 (*Figure 2A*) maintains apo-Mfd in its repressed state (*Deaconescu et al., 2006*; *Manelyte et al., 2010*; *Murphy et al., 2009*; *Smith et al., 2007*).

To visualize the expected conformational changes in Mfd upon EC engagement and de-repression, we analyzed the Mfd-EC complexes (*Figure 1*) by single particle cryo-EM. Steps of maximum-likelihood classification (*Scheres, 2012*) revealed seven Mfd-EC structures (L1, L2, C1-C5; *Figure 2B–H*) ranging from 3.4 to 4.1 Å nominal resolution (*Figure 2—figure supplement 1*; *Figure 2—figure supplement 2*; *Supplementary file 1*). With the exception of L1, the cryo-EM maps were of sufficient quality to directly observe bound nucleotide. Characteristics of the cryo-EM densities and of the surrounding protein structure led us to propose the nucleotide status of each structure (*Supplementary file 2*). We performed an objective, unbiased test of the proposed nucleotide identities by running identical refinements of each structure modeled with either ATP or ADP (aligned by their common atoms) and comparing the average real space correlation coefficients (*Adams et al., 2010*). The results (*Supplementary file 2*) suggested that C1, C2, and C5 contained either ATP or ADP•P (where the hydrolyzed γ-phosphate has not been released), and that C4 contained ADP (*Figure 2C–H*). Thus, Mfd was trapped progressing through its nucleotide hydrolysis cycle (NHC). According to our real space correlation coefficient test, the identity of the nucleotide bound to L2 and C3 was ambiguous based solely on the cryo-EM density (*Supplementary file 2*). We therefore denote the nucleotide status of these states with lowercase letter [i.e. L2(adp), C3 (adp)].

In all seven structures, the ECs have similar RNAP and nucleic acid conformations (root-mean-square-deviation, rmsd, for superimposed RNAP α-carbons; 0.59 Å < rmsd < 3.64 Å; *Supplementary file 3*. See Materials and methods for a description of the superimposition procedure. See *Figure 2—figure supplement 3A–G* for examples of cryo-EM density) and the Mfd-D4 (RID) maintains its interactions with the RNAP βprotrusion (*Figure 2B–H*; *Westblade et al., 2010*). By contrast, the conformation of Mfd and the disposition of the upstream duplex DNA vary dramatically (*Figure 2*, *Supplementary file 4*). As expected, all of the Mfd/DNA interactions occur through the DNA phosphate backbone.

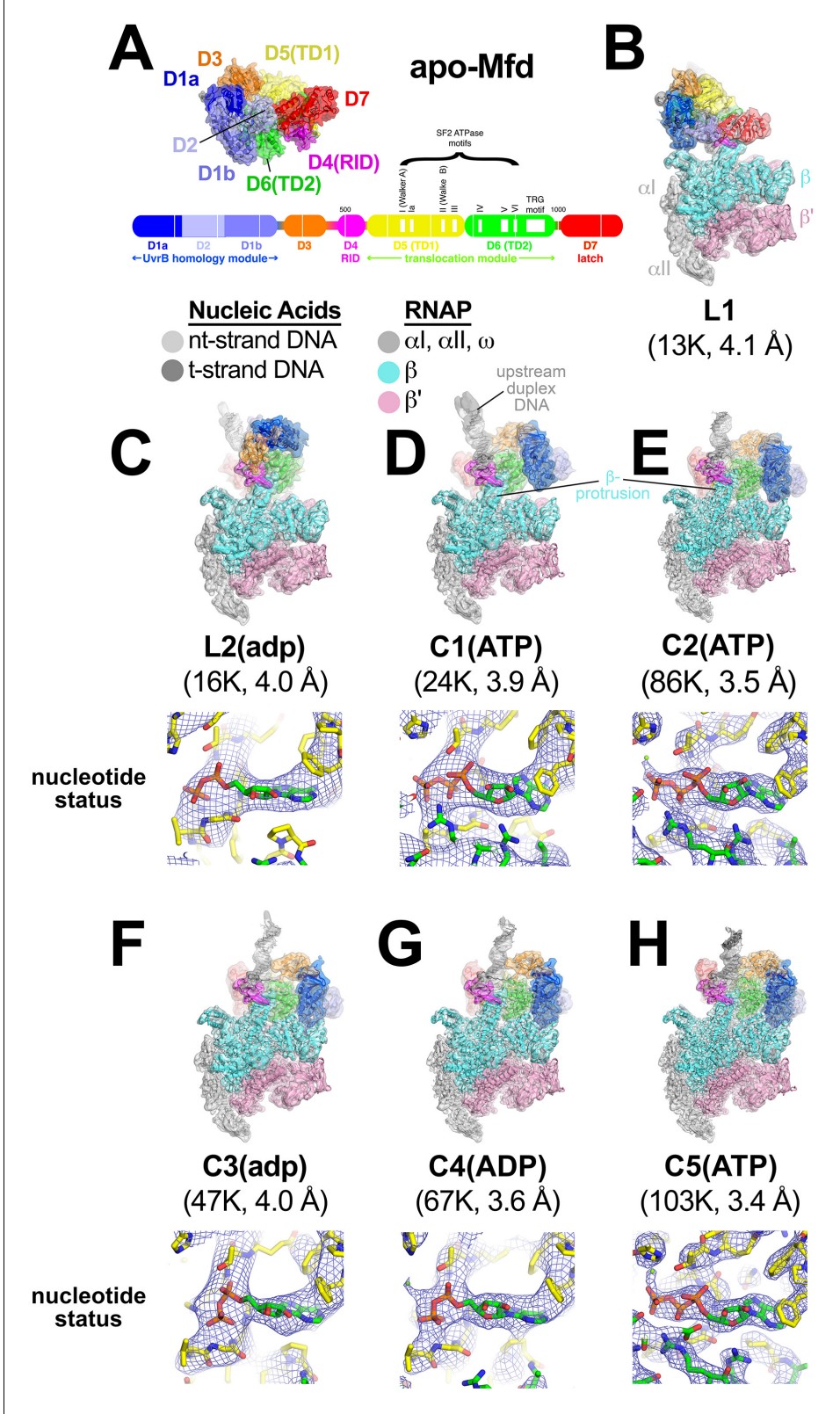

**Figure 2.** Mfd-elongation complex (EC) structures. (**A**) The structure of apo-Mfd [PDB 2EYQ; *Deaconescu et al., 2006*] is shown as a transparent molecular surface surrounding a backbone ribbon. The domain nomenclature and color coding are schematically represented by the horizontal bar below, which represents the 1148-residue *Eco* Mfd primary sequence (every 100 residues are marked by a vertical white line). Structural domains are shown as thick bars; thin bars represent connecting linkers. The UvrB homology module (D1a-D2-D1b) is structurally homologous to the namesake

*Figure 2 continued on next page*

*Figure 2 continued*

elements of UvrB (*Deaconescu et al., 2006*; *Selby and Sancar, 1993*). D4 is the RNA polymerase (RNAP) interacting domain (RID). D5 (Translocation Domain 1, or TD1) and D6 (TD2) contain the seven SF2 ATPase motifs denoted by white boxes and labeled (*Gorbalenya and Koonin, 1993*) as well as the TRG motif (*Chambers et al., 2003*; *Mahdi et al., 2003*), and together comprise the translocation module. (B)–(H). (*Top*) Overall structures of Mfd-EC complexes obtained by cryo-EM. The cryo-EM density maps low-pass filtered to the local resolution (*Cardone et al., 2013*) are shown as transparent surfaces with final models superimposed. Mfd is colored as shown in (A); the RNAP and nucleic acids are colored according to the key. (*Bottom*) Cryo-EM density (blue mesh) and superimposed models in the region around the Mfd nucleotide-binding site. Bound nucleotide could not be visualized in the L1 cryo-EM map (B) because of the low resolution. The nucleotide status (either ADP or ATP/ATP•P) could be determined from the cryo-EM map alone for C1(ATP), C2(ATP), C4(ADP), and C5(ATP) (see *Supplementary file 2*). Determination of the nucleotide status for L2(adp) and C3 (adp) was not possible from the cryo-EM maps alone (see *Supplementary file 2*), but other arguments suggest that these two states were bound to ADP (see text). (B) L1. (C) L2(adp). (D) C1(ATP). (E) C2(ATP). (F) C3(adp). (G) C4(ADP). (H) C5(ATP).

The online version of this article includes the following video and figure supplement(s) for figure 2:

**Figure supplement 1.** Cryo-EM processing pipeline for Mfd-elongation complex (EC) complexes.

**Figure supplement 2.** Cryo-EM of Mfd-elongation complex (EC) complexes.

**Figure supplement 3.** Examples of cryo-EM density and Mfd-induced DNA kink.

**Figure 2—video 1.** The Mfd loading cycle.

https://elifesciences.org/articles/62117#fig2video1

In most of the structures, the cryo-EM density in the region around the upstream edge of the transcription bubble is poor and difficult to interpret, likely due to heterogeneity in that region. As a consequence, the precise register of the upstream duplex DNA in the models is tentative. However, gross features of the upstream DNA, such as the paths of the DNA backbone and of the overall helical axes, are clear (*Figure 2—figure supplement 3A–G*). In this regard, the engagement of Mfd significantly distorts the upstream duplex DNA. Mfd induces a localized kink in the DNA, roughly centered in the footprint of the Mfd translocation module [D5(TD1)/D6(TD2)] on the DNA. The Mfd translocation module engages with the duplex DNA from the minor groove; the kink, which ranges from 7° to 15° away from Mfd, is accompanied by significant widening of the DNA minor groove (*Figure 2—figure supplement 3H*). The entire length of the upstream duplex DNAs bends away from the Mfd translocation module by 16–45° (*Figure 2—figure supplement 3A–G*). Based on single molecule observations, the Mfd interaction with DNA was proposed to induce bending or wrapping of the DNA (*Howan et al., 2012*), consistent with these structural observations.

## One ATP hydrolysis cycle corresponds to translocation by one base pair

Sequence analysis identifies Mfd as an SF2 ATPase (*Gorbalenya and Koonin, 1993*) with a RecA-type catalytic core that is most closely related to RecG (*Chambers et al., 2003*; *Mahdi et al., 2003*). Mfd and RecG are unique among SF2 ATPases in harboring a conserved 'TRG' (translocation in RecG) motif following the seven SF2 ATPase signature motifs. Mutations in conserved residues of the TRG motif uncouple ATP hydrolysis from duplex DNA translocation (*Chambers et al., 2003*; *Mahdi et al., 2003*). It has not been possible to understand the relationship between duplex DNA binding, the nucleotide status of the RecA catalytic core, and duplex DNA translocation mediated by the TRG motif due to the lack of structures of an SF2 translocase bound to duplex DNA in different nucleotide states. The series of Mfd-EC structures determined here help in this understanding.

The seven SF2 ATPase signature motifs cluster together at the interface between TD1 and TD2 where the nucleotide binds (*Figure 3A*). Since some of the structural states contain ATP (C1, C2, and C5) while others contain ADP (C4, and likely L2 and C3; *Figure 2B–H*), we can compare the disposition of the translocation domains with respect to each other, and correlate with the nucleotide status.

We superimposed α-carbons of residues 580–780 of TD1 (excluding the relay helix, which undergoes very large structural changes), yielding rmsds ranging between 0.282 and 0.894 Å (*Figure 3B*). The TD1 superimposition yielded two discrete positions of TD2, one of which corresponded with the ATP-bound structures (C1, C2, C5; green in *Figure 3B*), and the other with the ADP-bound structures (L2, C3, C4; blue in *Figure 3B*). L1, the complex in which the resolution of the cryo-EM map in the region of the Mfd nucleotide binding site (*Figure 2—figure supplement 2C*) was insufficient to

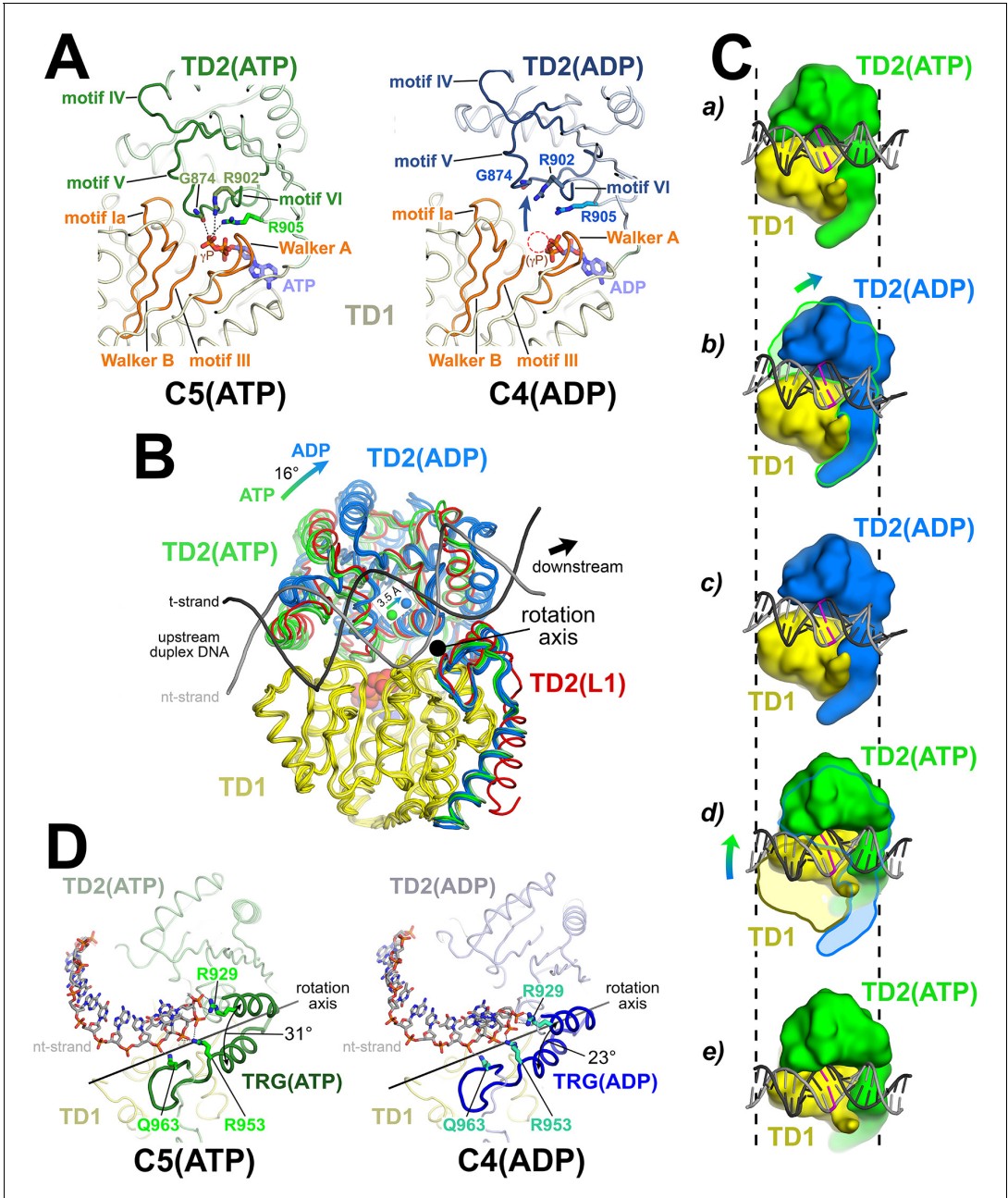

**Figure 3.** The Mfd translocation module and the DNA translocation mechanism. (**A**) Conformational changes of the Mfd translocation module induced by ATP hydrolysis and Pi release. The structural environments of the ATP [left; C5(ATP)] or ADP [right; C4(ADP)] binding sites are shown. The protein is shown as a backbone worm; TD1 is colored pale yellow but the SF2 ATPase motifs of TD1 (Walker A, motif Ia, Walker B, motif III) are colored orange; TD2(ATP) or (ADP) are colored pale green or light blue, respectively, but the SF2 ATPase motifs of TD2 (motifs IV, V, and VI) are colored dark green or dark blue. The nucleotide is shown in stick format with blue carbon atoms. The side chain or backbone atoms of three key residues, G874 (motif V), R902 (motif VI), and R905 (just beyond motif VI), are also shown. The backbone carbonyl of G874 and the side chains of R902 and R905 form polar interactions with the ATP γ-phosphate (denoted as gray dashed lines). In the ADP structure, these interactions are lost due to the missing γ-phosphate (denoted by the dashed red circle), causing TD2 to swing away from TD1 (denoted by the thick arrow). Here, 'polar contacts' include both hydrogen bonds (≤3.5 Å) and ionic interactions. Ionic interactions can include both close range interactions (where the hydration shells of the two oppositely charged moieties are displaced) that are typically called salt-bridges (≤3.5 Å), but can also include longer range interactions where the two oppositely charged moieties remain hydrated but their Coulombic interaction is favorable (***Kumar and Nussinov, 2002***; ***Xu et al., 1997***). These longer-range favorable interactions are significant and extend to well beyond 4.5 Å distance (***Yu et al., 2019***). (**B**) The translocation modules of all seven Mfd-elongation complex (EC) structures were superimposed by alignment of TD1 (colored yellow) α-carbons. The resulting positions of TD2 clustered into two groups, those with ATP (TD2 colored green) or ADP (TD2 colored blue). TD2 of L1 is shown in red and clusters with the ATP-bound structures. The

*Figure 3 continued on next page*

*Figure 3 continued*

relative disposition of the upstream duplex DNA is also shown (gray phosphate backbone worms). TD2(ATP) and TD2(ADP) are related by an ~16° rotation (denoted by the thick arrow, upper left) about an axis roughly perpendicular to the DNA helical axis (denoted by the black dot), resulting in a 3.5 Å shift of the TD2 center of mass roughly parallel to the DNA helical axis [center of mass positions for TD2(ATP) and TD2(ADP) denoted by the green and blue spheres, respectively], corresponding to one base pair rise of B-form DNA. The 3.5 Å shift of the TD2 center of mass is not sensitive to which structures are used to perform the calculation. (C) Inchworm model for duplex DNA translocation. Duplex DNA is shown as a cartoon (for reference, a central base pair is colored magenta). TD1 is colored yellow, while TD2 is colored green (ATP) or blue (ADP). In (a), both TD1 and TD2(ATP) interact with the duplex DNA (the initial positions of TD1 and TD2 on the DNA are denoted by the vertical dashed reference lines). Upon ATP hydrolysis and Pi release, TD2(ADP) rotates away from TD1 (b) and interacts with the DNA one base pair downstream (to the right, c). With the release of ADP, ATP binding induces TD1 to rotate toward TD2 (d). In (e), TD1 and TD2(ATP) both interact with the duplex DNA but one base pair to the right. Also see *Video 1*. (D) Conformational changes of the TRG motif. Protein is shown as a backbone worm; TD1 is colored pale yellow; TD2(ATP) or (ADP) are colored pale green or light blue, respectively, but the TRG motifs are colored dark green or dark blue. The nt-strand of the upstream duplex DNA is shown in stick format (the t-strand of the DNA is not shown for clarity). Three key TRG motif residues interact with the nt-strand DNA backbone, R929, R953, and Q963 (side chains shown, polar interactions with the DNA denoted by the gray dashed lines). The rotation axis of the TD2(ATP) → TD2(ADP) conformational change passes directly through the TRG motif helical hairpin linker, which serves as the hinge. Opening of TD2(ADP) causes the TRG helical hairpin to pinch closed nearly 10 Å.

directly assign the nucleotide status, clearly groups with the ATP-bound structures (red in *Figure 3B*). Previous studies have established that Mfd does not stably interact with DNA in the absence of ATP or a non-hydrolyzable ATP analog such as ATP-γ-S (*Chambers et al., 2003*; *Howan et al., 2012*; *Selby and Sancar, 1995a*; *Selby and Sancar, 1995b*; *Smith et al., 2007*). Thus: (i) the translocation module of L1-Mfd is engaged with the DNA (see below), and so is presumably bound to ATP, and (ii) the conformational state of the L1-Mfd translocation module groups with the other ATP-bound structures, and we infer that L1 contains Mfd(ATP). As with L2(adp) and C3 (adp), since we do not directly observe ATP bound to L1-Mfd from the cryo-EM density, we denote the state as L1(atp).

Mfd amino acid side chain interactions with the bound nucleotide and with the DNA are clearly observed in the cryo-EM maps of C2(ATP), C5(ATP), and C4(ADP). Some of the side chain interactions discussed below are also directly observed in the C3(adp) cryo-EM maps, but many side chains lack clear density. Because of the low resolution of the Mfd region of the cryo-EM maps of L1(atp), L2(adp), and C1(atp), cryo-EM density for Mfd side chains is generally not observed for these states (*Figure 2—figure supplement 2*). Based on interactions between conserved residues and the bound nucleotide or with the DNA observed in the cryo-EM maps of C2(ATP), C4(ADP), and C5(ATP), we infer that similar interactions occur in the other states with lower resolution cryo-EM maps.

In the ATP-bound structures, the carbonyl oxygen of G874 (motif V), and the side chains of R902 (motif VI) and R905 (just beyond motif VI), all in TD2, form polar interactions either hydrogen bonds (≤3.5 Å) or longer-range ion-pair interactions (≤4.5 Å); see *Figure 3A* legend; (*Kumar and Nussinov, 2002*; *Xu et al., 1997*; *Yu et al., 2019*) with the ATP γ-phosphate (*Figure 3A*, left). These three residues are absolutely conserved in an alignment of 65 Mfd sequences (*Deaconescu et al., 2006*). In the absence of the γ-phosphate in the ADP structures, these interactions are lost and TD2 rotates away from TD1 [*Figure 3A* (right), 3B]. The movement of TD2 with respect to TD1 on transitioning from the ATP- to the ADP-bound state

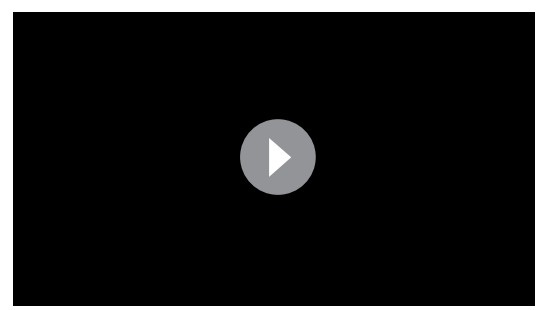

**Video 1.** Mfd translocation. The video illustrates the Mfd translocation module inchworm translocation model. The Mfd translocation module [D5(TD1), yellow; D6(TD2), green] transitions between the closed ATP-bound state and the open ADP-bound state as it translocates on duplex DNA. In the first section of the video, the Mfd translocation module inchworms on stationary DNA. The second section of the video simulates the effect of blocking the forward (left-to-right) motion of the protein but with continued ATP hydrolysis – the protein continues to translocate with respect to the DNA, but since the protein cannot move, the DNA moves (from right-to-left) instead. A reference base pair in the DNA is colored magenta.
https://elifesciences.org/articles/62117#video1

corresponds to an ~16° rotation about an axis roughly perpendicular to the helical axis of the upstream duplex DNA (*Figure 3B*). As a result, the center of mass of TD2 shifts parallel to the DNA helical axis about 3.5 Å in the downstream direction (*Figure 3B*), corresponding to one base pair rise of B-form DNA. These observations are suggestive of an 'inchworm' model for translocation, as proposed for related SF1 helicase translocation on single-stranded nucleic acids (*Lohman et al., 2008*; *Singleton et al., 2007*; *Velankar et al., 1999*; *Yarranton and Gefter, 1979*) and for Mfd based on single-molecule observations (*Le et al., 2018*). In this model (schematized in *Figure 3C*), upon hydrolysis and release of the γ-phosphate, TD2 moves forward on the duplex DNA and forms tight interactions one base pair downstream. Next, the TD1/DNA interactions loosen and ADP must exchange for ATP in the nucleotide-binding site. This allows TD1 to close toward TD2, reestablishing interactions with the ATP γ-phosphate, now with both TD1 and TD2 translocated one base pair downstream on the duplex DNA (*Figure 3C*; *Video 1*). The model predicts a translocation stepsize of one nucleotide per ATP hydrolysis cycle, consistent with measurements for SF1 helicases (*Dillingham et al., 2000*; *Tomko et al., 2007*).

The TRG motif, which couples ATP hydrolysis to translocation, contains a helical hairpin motif followed by a meandering loop structure (*Figure 3D*). Each of these structural elements harbors one of the three conserved TRG residues that are critical for translocation: R929 (1st helix), R953 (2nd helix), and Q963 (loop) (*Chambers et al., 2003*; *Mahdi et al., 2003*). All three of these residues interact with the DNA phosphate backbone. The rotation axis of the TD2(ATP) → TD2(ADP) transition passes directly through the linker connecting the helical hairpin (*Figure 3D*). Examining the structures more closely, the structural elements of TD2 C-terminal to the linker, which includes the second TRG hairpin-helix, the TRG loop, and the hook-helix, make extensive interactions with TD1 and move with TD1 as a rigid body, while the N-terminal portion of TD2 (residues 781–939) opens and closes depending on the nucleotide status. Thus, the linker connecting the TRG helical hairpin acts as the hinge (centered near absolutely conserved G942) for the TD1/TD2 conformational change in response to nucleotide status. Opening of TD2 in the ADP state results in closing of the TRG helical-hairpin (*Figure 3D*).

## Pathway for Mfd function

For the purpose of analyzing and discussing the structures, it is useful to place them in an ordered pathway. To begin the pathway, apo-Mfd from solution (*Figure 2A*) interacts with an EC. Comparing the Mfd component of each complex with the apo-Mfd structure (*Deaconescu et al., 2006*), only L1 (atp) has an rmsd <10 Å (all the others are >>30 Å; *Supplementary file 4*). Therefore we place L1 (atp) as the first structure in the pathway (*Figure 4—figure supplement 1A*).

To order the rest of the structures, we superimposed α-carbons of each complete structure (Mfd and RNAP α-carbons) with α-carbons of every other structure and calculated the rmsd of α-carbon positions, generating a table of 21 pair-wise rmsd values (*Supplementary file 5*). L1(atp) and L2 (adp) clearly stand apart from the other structures; L1 and L2 compared with every other structure exhibit rmsds between 19 Å and 47 Å, while C1–C5 compared with each other exhibit rmsds between 3.7 Å and 11.4 Å. We therefore propose that L1(atp) and L2(adp) are Mfd 'Loading' complexes, while C1(ATP), C2(ATP), C3(adp), C4(ADP), and C5(ATP) are related structures looping in the Mfd NHC. Starting with L1(atp), the path of transitions from one structure to the next that gives the smallest cumulative rmsd (*Supplementary file 5*) is shown in *Figure 4—figure supplement 1A*.

## Mfd loading requires multiple rounds of ATP hydrolysis and is accompanied by profound conformational changes

The Mfd component of L1(atp) [Mfd(atp)$_{L1}$], the first experimental structure in the pathway (*Figure 4—figure supplement 1A*), is most similar to the structure of apo-Mfd [PDB 2EYQ; (*Deaconescu et al., 2006*)]. We modeled a presumed precursor to L1, [L0] (square brackets denote a structural model), by superimposing the apo-Mfd-D4(RID) structure onto the Mfd(atp)$_{L1}$-D4(RID) (*Figure 4A*). This reveals that the [L0] → L1 transition involves large translations and rotations of TD1 (11 Å translation, 43° rotation) and TD2 (16 Å translation, 37° rotation; *Supplementary file 5*), bringing the two ATPase domains into alignment, presumably to bind ATP and engage with the DNA (*Figure 4B,C*).

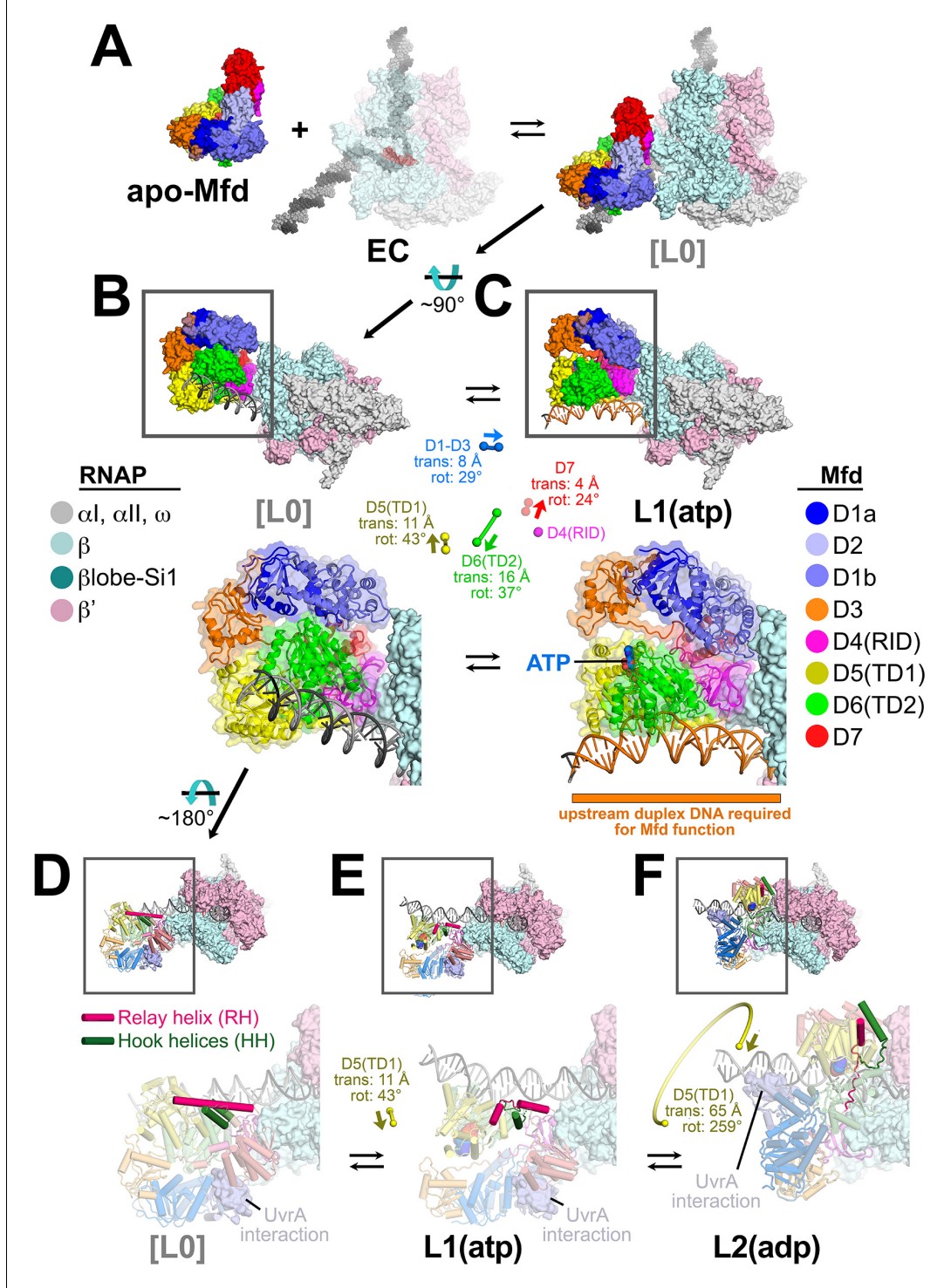

**Figure 4.** Initial stages of the Mfd loading pathway. Color coding of RNA polymerase (RNAP) subunits and Mfd domains are shown in the keys on the left and right, respectively. (**A**) Apo-Mfd PDB 2EYQ (**Deaconescu et al., 2006**) combines with an elongation complex (EC) [PDB 6ALF (**Kang et al., 2017**) with upstream and downstream duplex DNA extended] to form a putative initial encounter complex [L0], which was modeled by superimposing apo-Mfd D4(RID) onto the Mfd$_{L1}$-D4(RID) and adjusting the trajectory of the upstream duplex DNA. (**C**) The [L0] → L1(atp) transition is shown. In this view, the downstream duplex DNA (and the direction of transcription) points away from the viewer. (*Top*) The Mfd-EC structures are shown as molecular surfaces with DNA shown in cartoon format. The boxed regions are magnified below. (*Bottom*) Mfd is shown as a transparent molecular surface surrounding a backbone ribbon. In the middle, the colored spheres denote the relative positions of the Mfd domain center-of-masses (com), with connecting lines denoting the motions from the L[0] → L1(atp) transition (the translations of the com's, as well as the relative rotation of the domains,

*Figure 4 continued on next page*

*Figure 4 continued*

are listed. The D4(RID) motion is negligible; also see *Supplementary file 4*). D.[L0]. E.L1(atp): The region of the upstream duplex DNA colored orange and denoted by the orange stripe was found to be required for Mfd function on an EC (*Park et al., 2002*). (F) View of the [L0] → L1(atp) → L2(adp) transition, highlighting the structural changes in the Mfd relay helix (RH) and hook helices (HH). In this view, the Mfd-EC complex is rotated ~180° about a horizontal axis, so the downstream duplex DNA (and direction of transcription) is toward the viewer. (*Top*) The RNAP is shown as a molecular surface, with nucleic acids shown in cartoon format. Mfd is shown with cylindrical helices. Color coding is as above but the RH is colored hot pink, and the HHs are colored dark green. The boxed region is magnified below. (*Bottom*) The complexes are shown in faded colors except for the RH and HHs. Also shown as a molecular surface are the residues of Mfd-D2 that interact with UvrA [determined from PDB 4DFC (*Deaconescu et al., 2012*)]. D.L[0]: The RH at the very N-terminus of TD1 extends for 30 residues and is surrounded by the HHs at the very C-terminus of TD2. The UvrA-interacting surface of Mfd-D2 is occluded by D7 (*Deaconescu et al., 2006*). E.L1(atp): The middle portion of the RH helix unfolds and the RH kinks about 112° around the second HH due to the translation/rotation of TD1 (denoted) and also TD2. The UvrA-interacting surface of Mfd-D2 is still occluded by D7. F.L2(adp): The transition from L1(atp) → L2(adp) involves a 259° rotation of TD1 around the backside of the DNA, as well as a 65 Å translation toward the RNAP (denoted). This is likely accomplished by ATP-hydrolysis-dependent walking of the Mfd translocation module and D7 along the DNA until it bumps into the RNAP. The corkscrewing translocation module unfolds the N-terminal half of the RH, wrapping it around the DNA as it goes. In this process D2 is separated from D7 but the UvrA-interacting surface of D2 is now occluded by the DNA. Also see *Figure 2—video 1*.

The online version of this article includes the following figure supplement(s) for figure 4:

**Figure supplement 1.** Putative structural pathway for Mfd activity and Mfd/DNA interactions.

Mfd requires ~30 base pairs of duplex DNA upstream of the EC transcription bubble [to about −40 (*Park et al., 2002*)]. Mfd/DNA interactions occur between about −34 and −14 with one exception, Mfd(atp)$_{L1}$ interacts with the DNA further upstream, from about −38 to −27 (*Figure 4C*, *Figure 4—figure supplement 1B*), explaining the result of *Park et al., 2002* and also confirming that L1(atp) is an obligate intermediate in the Mfd loading pathway. The Mfd translocation module makes extensive interactions with both DNA strands in each structure, but the direct interactions slightly favor the t-strand DNA; about 60% of the direct Mfd-translocation module:DNA interactions occur with the t-strand DNA in each of the structures.

In apo-Mfd, The D4(RID) is connected to the first RecA ATPase domain (TD1) by a 30-residue α-helix, the Relay Helix (RH, residues 548–577; *Figure 4D*). The RH at the N-terminus of TD1 interacts with the hook helices at the very C-terminus of TD2. In the [L0] → L1(atp) transition, the translations and rotations of TD1 and TD2 result in unfolding of seven residues in the middle of the RH (561–567), and kinking of the RH ~112° around the second hook helix. The first hook helix also completely unfolds (*Figure 4E*).

The transition from L1(atp) → L2(adp) involves remarkable rearrangements of the Mfd structural modules. Other than the D4(RID), which stays anchored to the RNAP βprotrusion, the minimum center-of-gravity translation of an Mfd structural module [D1–D3, D5(TD1), D6(TD2), D7] is 60 Å, while the minimum rotation is 148° (*Figures 4F* and *5*; *Supplementary file 6*). As a result of the large conformational rearrangement of Mfd, Mfd(adp)$_{L2}$ is topologically 'wrapped' around the DNA (*Figure 5*), likely explaining how Mfd (with RNAP in tow) translocates processively over many kilobases of DNA (*Fan et al., 2016*; *Graves et al., 2015*).

Although it is difficult to imagine the choreography of the Mfd structural modules in the L1(atp) → L2(adp) transition without parts of Mfd passing through itself or through the DNA, a pathway exists. First, D1–D3 must dissociate from its position in L1(atp), generating a hypothetical intermediate [L1.5a] (*Figure 5*). The movement of D1–D3 could be triggered by initial rounds of ATP hydrolysis/translocation by the translocation module and is facilitated by the 25 amino acid linker connecting D3 with D4(RID) (*Figures 2A* and *5*). The release of D1–D3 now opens a path for the Mfd translocation module to 'walk' along the DNA, corkscrewing in the downstream direction (clockwise in the view of *Figure 5*) until it bumps into the RNAP at its position in L2 (*Figures 4F* and *5*). As the Mfd translocation module corkscrews along the DNA, the N-terminal part of the RH (residues 548–560) completely unfolds and is dragged around the DNA, forming part of the topological link of Mfd on the DNA (*Figure 4F*). This proposed path of conformational changes is most easily understood by viewing *Figure 2—video 1*.

The initial engagement of Mfd with the RNAP [in the putative [L0] or in L1(atp)] is through the Mfd-D4(RID):βprotrusion interaction [average interface area of 553 Å$^2$, calculated using the PDBe-PISA server; (*Krissinel and Henrick, 2007*)] and this interaction does not change through all seven

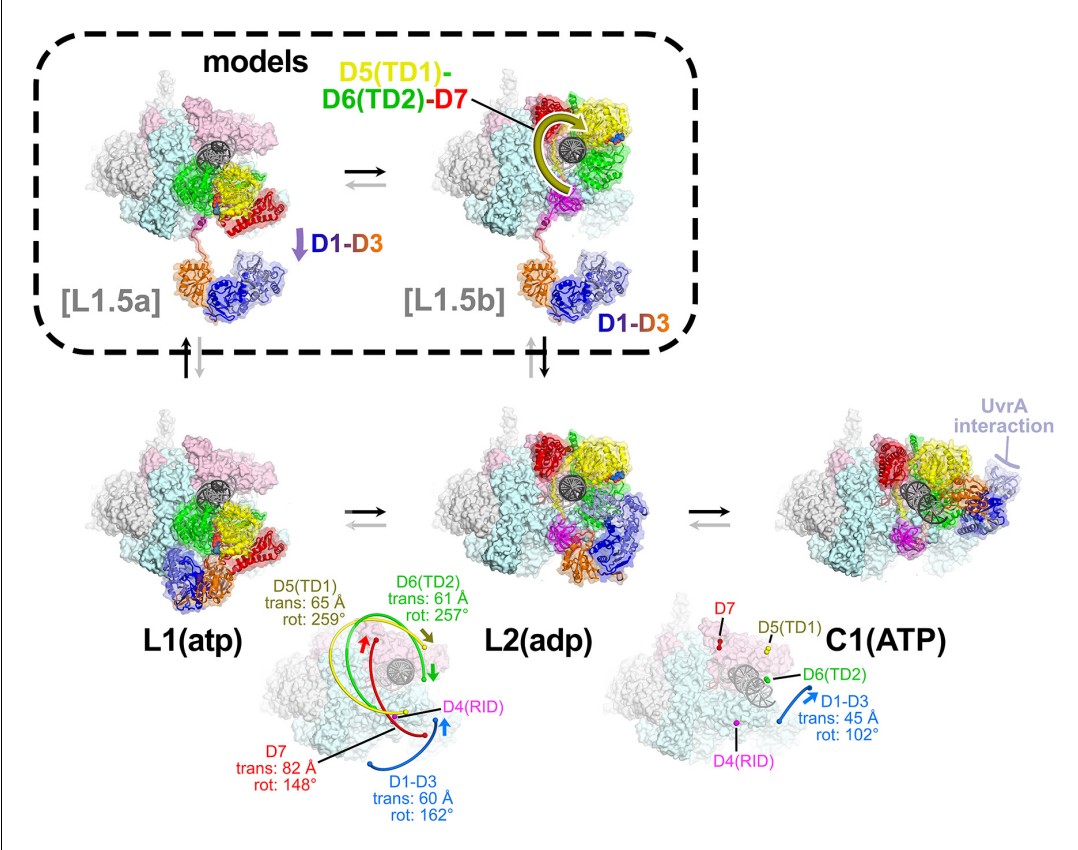

**Figure 5.** The L1(atp) → L2(adp) → C1(atp) transition. (**A**) A back view of the Mfd-elongation complex (EC) structures is shown, viewing down the axis of the upstream duplex DNA (the direction of transcription would be to the right). The RNA polymerase (RNAP) is shown as a molecular surface with nucleic acids shown in cartoon format. Mfd is shown as transparent molecular surfaces surrounding the backbone ribbon. The bottom row shows the transition through the experimental structures [L1(atp) → L2(adp) → C1(ATP)]. Below that is denoted the translations and rotations of the Mfd domains associated with each transition (superimposed on the EC structure). The large translation/rotation of Mfd D5(TD1)-D6(TD2)-D7 in the L1(atp) → L2(adp) transition must occur by clockwise corkscrewing around the DNA minor groove in order to leave behind the unfolded portion of the RH, which wraps around the DNA. The configuration of Mfd D1–D3 in L1(atp) would block this transition and also result in the entanglement of linkers; thus, we propose that this transition is facilitated by transient intermediates [L1.5a] and [L1.5b], which have been modeled with displaced Mfd D1–D3 tethered by the long linker connecting D3 with the D4(RID). This allows the unencumbered transition of Mfd D5(TD1)-D6(TD2)-D7 from [L1.5a] to [L1.5b] (illustrated by the thick yellow arrow). Mfd D1-D3 then accommodates in a new configuration in L2(adp). The L2(adp) → C1(ATP) transition involves another large translation and rotation of D1–D3 (denoted), which finally exposes the Mfd-D2 UvrA-interacting surface. This transition is also illustrated in *Figure 2—video 1*.

The online version of this article includes the following figure supplement(s) for figure 5:

**Figure supplement 1.** Selected interdomain Mfd:RNAP interface areas.

states (L1, L2, and C1–C5; *Figure 5—figure supplement 1*). In L1, the Mfd translocation module [D5(TD1)/D6(TD2)] interacts with upstream DNA (roughly −38 to −27; *Figure 4—figure supplement 1B*) and does not interact with RNAP (*Figure 5—figure supplement 1*). Upon the transition to L2, the translocation module walks on the DNA, moving toward the RNAP until it 'bumps' into the RNAP, characterized primarily by D6(TD2) interacting with the RNAP βprotrusion [the Mfd-D4(RID) and D6(TD2) interfaces with the RNAP βprotrusion do not overlap; *Figure 5—figure supplement 1*]. Once Mfd-D6(TD2) pushes up against the RNAP βprotrusion, that interface is also maintained throughout the rest of the structures (L2 and C1–C5, average interface area of 312 Å$^2$; *Figure 5—figure supplement 1*).

In L1(atp), the UvrA-interacting surface of D2 is occluded through its interaction with D7, as it is in apo-Mfd [*Figure 4D,E*; *Deaconescu et al., 2006*; *Deaconescu et al., 2012*]. During the large rearrangements in the L1(atp) → L2(adp) transition, D7 travels with the translocation module, separating

it from D2 (*Figure 4F*). However, in L2(adp), the UvrA-interacting surface of D2 is still occluded by the DNA (*Figure 4F*).

The L2(adp) → C1(ATP) transition involves another large rearrangement of D1–D3 (45 Å translation, 102° rotation; *Figure 5*; *Supplementary file 5*), which accommodates into its position seen in C1(ATP) → C5(ATP). In this configuration, the UvrA-interacting surface of D2 is finally exposed to solution (*Figure 5*). The other Mfd structural modules make relatively small motions and Mfd remains topologically wrapped around the DNA (*Figure 5*).

The alternating nucleotide states on the transition from L1(atp) → L2(adp) → C1(ATP) suggest that the complete loading of Mfd involves rounds of ATP hydrolysis. Furthermore, since the translocation module traverses nine base pairs on the DNA in the L1(atp) → L2(adp) transition, at least nine molecules of ATP must be hydrolyzed (*Figure 2—video 1*; *Figure 4—figure supplement 1A*). Note this is before Mfd has entered the NHC that serves to displace the RNAP (*Figure 4—figure supplement 1A*).

To test the structure-based hypothesis that ATP hydrolysis is required for Mfd loading to achieve a stable Mfd-EC complex, we incubated ECs and Mfd with vanadate ($VO_4^{3-}$) and either ADP or ATP. With ADP in the nucleotide-binding site, vanadate can bind in the position normally occupied by the γ-phosphate; the $ADP-VO_4$ complex is thought to mimic the ATP hydrolysis transition state and is an effective inhibitor of ATP binding and hydrolysis (*Davies and Hol, 2004*). ADP and vanadate from solution bind directly in the nucleotide-binding site without any rounds of ATP hydrolysis since no ATP is present. With ATP and vanadate, on the other hand, at least one round of ATP hydrolysis can occur. Following ATP hydrolysis, vanadate substitutes for the leaving inorganic phosphate before ADP can be released, inhibiting further ATP hydrolysis (*Oldham and Chen, 2011*; *Shimizu and Johnson, 1983*).

Recall that incubating Mfd and ATP with ECs formed on a nucleic acid scaffold containing a partially non-complementary transcription bubble gave rise to a unique complex observed by EMSA (*Figure 1*, 5B, band labeled '*' in lane 1). ADP + vanadate does not support complex formation, while ATP + vanadate does (*Figure 6A*, lanes 2 and 3). The same concentrations of ATP + vanadate completely inhibited Mfd function in an EC displacement assay (*Chambers et al., 2003*; *Figure 6B*). These experiments establish that at least one round of ATP hydrolysis is required for Mfd to form a stable complex with an EC.

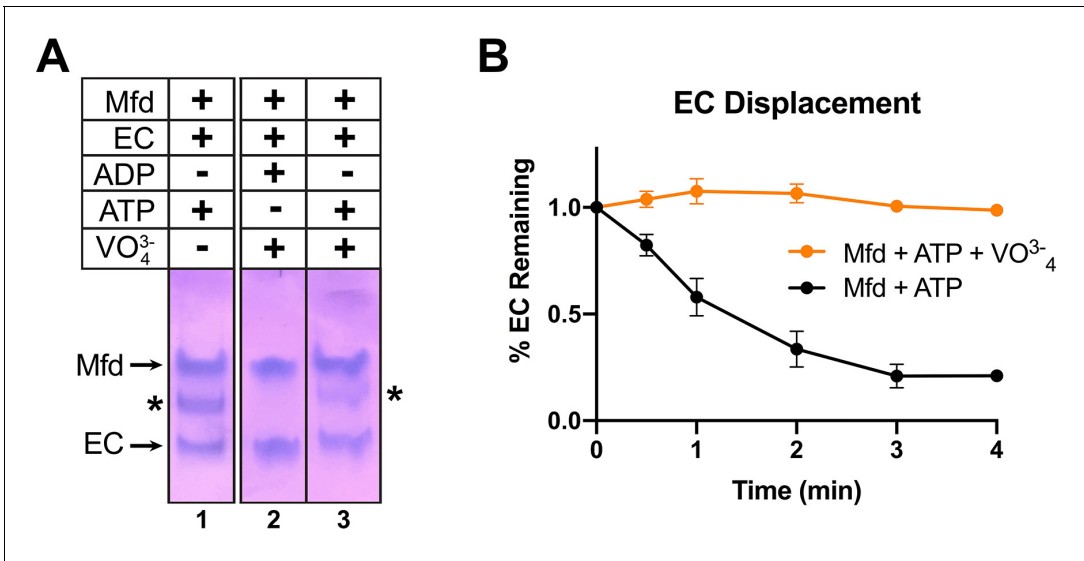

**Figure 6.** ATP hydrolysis is required for Mfd loading. (A) Native gel electrophoretic mobility shift assay shows that a stable Mfd-elongation complex (EC) complex (denoted by '*') requires a minimum of one round of ATP hydrolysis (lane 3). (B) Displacement of stalled *Eco* ECs from end-labeled DNA fragments by Mfd was monitored by electrophoretic mobility shift analysis and quantified using a phosphorimager and Imagequant software (*Chambers et al., 2003*). Data shown are the average of three independent experiments and are expressed as a percentage of the amount of EC present prior to the addition of Mfd. Error bars indicate standard deviation of three independent measurements.

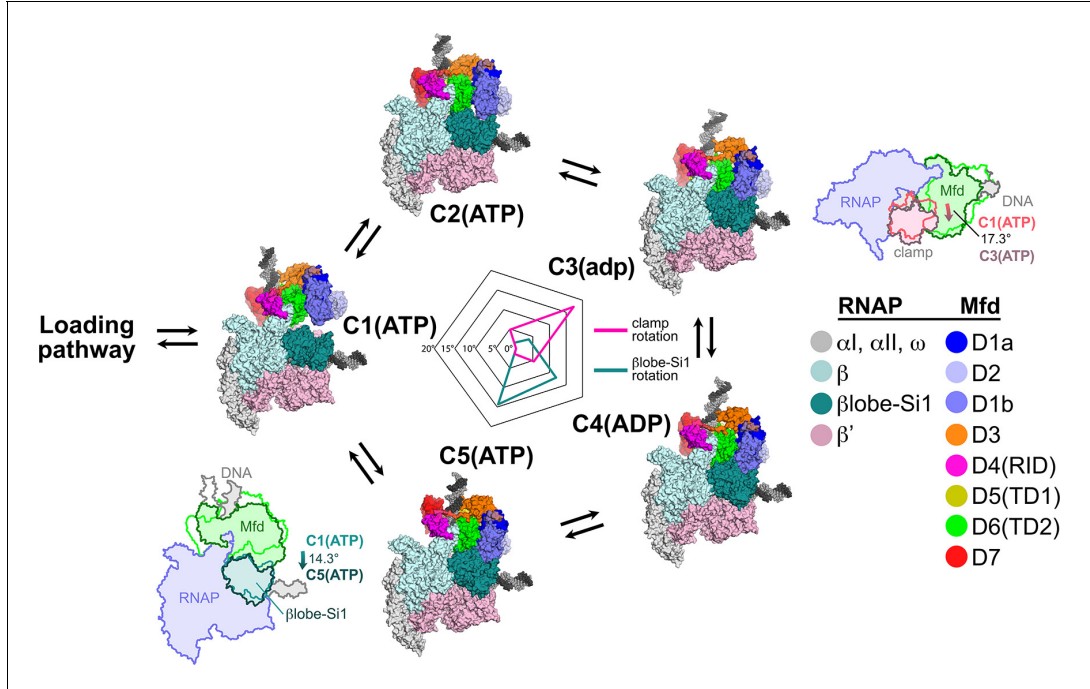

**Figure 7.** Mfd motions during its nucleotide hydrolysis cycle cause significant RNA polymerase (RNAP) conformational changes. The completion of the Mfd loading pathway culminates in the formation of C1(ATP) (*Supplementary file 5*). Mfd then cycles through five distinct states in the order proposed here (also see *Supplementary file 5*, *Figure 4—figure supplement 1A*, and *Figure 7—video 1*). In looping through this cycle, internal conformational changes of Mfd are relatively small (involving primarily the nucleotide-dependent shifts of D5(TD1) and D6(TD2) with respect to each other; see *Figure 3*), but Mfd and the upstream duplex DNA as a whole wobble back and forth by about 30° on the upstream face of the RNAP. These motions cause significant RNAP conformational changes quantified in the radar plot in the middle. Using C1(ATP) as a reference structure, the RNAP clamp of C3(adp) is opened 17.3° (schematically illustrated in the cartoon inset). The βlobe-Si1 domain of C5(ATP) is rotated 14.3° as illustrated.

The online version of this article includes the following video for figure 7:

**Figure 7—video 1.** The Mfd nucleotide hydrolysis cycle.

https://elifesciences.org/articles/62117#fig7video1

## Mfd manipulates the RNAP conformation during its NHC

After the Mfd loading pathway results in the formation of C1(ATP), we propose that Mfd then loops through an ATP hydrolysis cycle, captured in the five distinct structural states [C1(ATP) ↔ C2(ATP) ↔ C3(adp) ↔ C4(ADP) ↔ C5(ATP) ↔ C1(ATP)...; *Figure 7*]. Compared to the profound conformational changes that occur through the loading pathway (*Figures 4* and *5*), the conformations of Mfd in C1–C5 are similar to each other. Comparing the Mfd loading conformations of L1(atp), L2(adp), and C1(ATP) gives rmsd values ranging from 26.0 Å to 38.4 Å, while comparing C1(ATP) through C5 (ATP) gives rmsd values ranging from 3.95 Å to 7.68 Å (*Supplementary file 4*). Although the Mfd conformations in C1(ATP) through C5(ATP) are similar, Mfd as a whole, along with the upstream duplex DNA, wobbles back and forth with respect to the EC by more than 30° (*Figure 7*, *Supplementary file 6*).

During its NHC (C1–C2–C3–C4–C5–C1...), Mfd translocates on the upstream duplex DNA in the downstream direction, tracking in the minor groove (*Figure 3*, *Video 1*). If the RNAP is unable to translocate forward, this results in overwinding of the upstream DNA, facilitating collapse of the transcription bubble and release of the RNA transcript (*Park and Roberts, 2006*). However, the motions of Mfd during the NHC also induce significant conformational changes to the RNAP itself: clamp opening and βlobe-Si1 rotation (*Figure 7*).

The conformation of RNAP in C1(ATP) is very similar to the conformation of RNAP in an EC (*Kang et al., 2017*). Using C1(ATP) as a reference conformation for RNAP (clamp opening = βlobe-Si1 rotation = 0°), the conformational changes of Mfd during its NHC cause opening and

closing of the clamp, with the peak of clamp opening, 17.3°, at C3(adp) (*Figure 7*). Similar magnitudes of clamp opening have been observed in many structures, without (*Zhang et al., 1999*) and with nucleic acids (*Tagami et al., 2010*; *Weixlbaumer et al., 2013*).

The Mfd conformational changes through the NHC also cause a sideways rotation of the βlobe-Si1 module, with the peak of βlobe-Si1 rotation, 14.3°, at C5(ATP) (*Figure 7*). The direction and magnitude of the βlobe-Si1 rotation are very similar to an RNAP conformational change induced by TraR binding (*Chen et al., 2019a*). In the case of TraR, this conformational change occurs in the context of an initiation complex with promoter DNA and has multiple effects depending on the promoter context (*Chen et al., 2019a*). In the context of an EC, we propose that the βlobe-Si1 rotation alters the RNAP-downstream DNA duplex contacts in the RNAP cleft, destabilizing these interactions and thereby destabilizing the EC (*Nudler et al., 1996*). The conformational transitions during the Mfd-NHC are visualized in *Figure 7—video 1*.

During the Mfd-NHC, the movements of Mfd relative to the RNAP cause cyclical RNAP conformational changes involving RNAP clamp and βlobe motions (*Figure 7*). The RNAP clamp is opened in C3 as Mfd wedges itself between the βprotrusion and the clamp, pushing on the clamp through a large interface with the translocation module (maximum Mfd-[D5(TD1)/D6(TD2)]:β'clamp interface area of 944 Å$^2$ in C3; *Figure 5—figure supplement 1*). The RNAP βlobe-Si1 is pushed sideways by an interaction with Mfd-D1 (can be seen in *Figure 7*; *Figure 5—figure supplement 1*).

## Discussion

The crystal structure of apo-Mfd revealed a compact but complex arrangement of structural modules connected by long, flexible linkers [*Figure 2A*; *Deaconescu et al., 2006*]. Interdomain interactions maintain Mfd in a repressed state, where its activities of ATP hydrolysis, DNA translocation, and UvrA binding are suppressed (*Manelyte et al., 2010*; *Murphy et al., 2009*; *Smith et al., 2007*). Amino acid substitutions that disrupt key interdomain interactions cause profound conformational changes in Mfd and unleash its ATPase and DNA translocase activities (*Manelyte et al., 2010*; *Murphy et al., 2009*; *Smith et al., 2007*). In this work, we observed seven distinct structures (*Figure 2B–H*) that delineate how the initial interaction of Mfd with an EC triggers a stepwise series of dynamic conformational changes, culminating in the stable engagement of Mfd with the EC and then ATP-hydrolysis powered disruption of the EC. Key structural features and conformational changes are highlighted in the context of the transition path from one structure to the next (*Figure 4—figure supplement 1A*) in the summary *Figure 8*.

Initial binding of the Mfd-RID to the RNAP βprotrusion to generate the hypothetical state [L0] (*Figures 4A,B* and *8*) does not appear to require conformational changes as presumed, but rather tethers the Mfd translocation module in close proximity to the upstream duplex DNA of the EC (*Figure 4B*). The disposition of TD1 and TD2 in [L0] is not conducive to DNA or ATP binding (*Deaconescu et al., 2006*), but thermal breathing of the Mfd domains could transiently align TD1 and TD2 to allow ATP binding and stable engagement of the upstream duplex DNA in L1(atp) (*Figures 4C* and *8*). In L1(atp), Mfd engages with the upstream duplex DNA to −38 (*Figure 4—figure supplement 1B*), explaining why Mfd requires upstream duplex DNA to about −40 to displace the EC (*Park et al., 2002*). Although our modeled pathway initiates with apo-Mfd interacting directly from solution with the stalled EC to generate [L0] and then L1 (*Figure 4A–C*), our results do not rule out a role for the 'catch and release' model of *Le et al., 2018* in which Mfd translocates on duplex DNA on its own and can engage with a stalled EC if encountered.

The alignment of TD1 and TD2 and ATP binding allows ATP hydrolysis in L1(atp), which initiates inchworming of the Mfd translocation module in the downstream direction (*Figure 3*; *Video 1*). We hypothesize that this induces the displacement of D1–D3 (which is still tethered), clearing a path for continued translocation of TD1/TD2 (*Figure 5A*, *Figure 2—video 1*). TD1/TD2 walk along the duplex DNA (*Figure 3*), corkscrewing around the DNA for nine base pairs and in the process leave the unfolded relay-helix polypeptide wrapped around the DNA in L2(adp) (*Figures 4F* and *5*, *Figure 2—video 1*). This ATP-hydrolysis-driven choreography results in Mfd completely encircling the upstream duplex DNA, explaining the remarkable processivity of the translocating Mfd-RNAP complex subsequent to EC disruption (*Fan et al., 2016*; *Graves et al., 2015*).

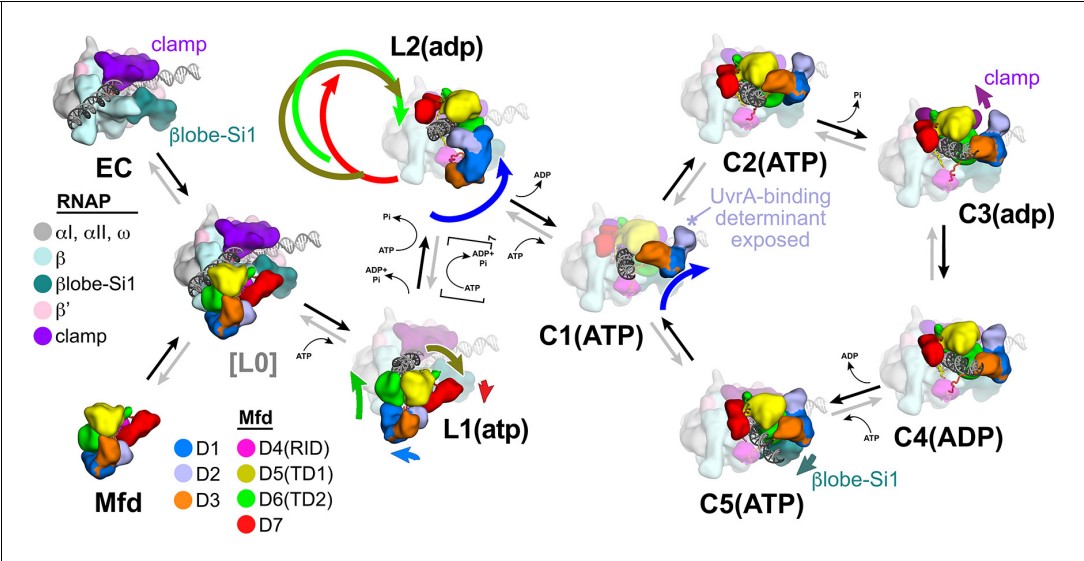

**Figure 8.** Structural pathway for RNA polymerase (RNAP) elongation complex (EC) displacement by Mfd. Putative ordered pathway for Mfd-mediated displacement of the RNAP EC (see *Figure 4—figure supplement 1A*). Structures are shown as cartoons, with RNAP and Mfd-domain color coding shown in the keys on the left. For most of the structures, domains that undergo the most significant conformational changes at each step are highlighted, and the relative direction and magnitude of the conformational changes are indicated by the thick, colored arrows. Starting at the left, the RNAP EC (top left) and apo-Mfd (bottom left) combine to form the putative encounter complex [L0] (see *Figure 4A,B*). In the [L0] → L1(atp) transition (*Figure 4B*), Mfd domains D5(TD1) and D6(TD2) rearrange to bind a molecule of ATP and engage fully with the upstream duplex DNA. D1–D3 and D7 also make small rearrangements. In the L1(atp) → L2(adp) transition, the Mfd translocation module [D5(TD1)/D6(TD2)] walks nine base pairs in the downstream direction (into the page) by clockwise corkscrewing around the duplex DNA approximately 260°, hydrolyzing nine molecules of ATP in the process. The transition to C1(ATP) involves another accommodation of D1–D3 which finally exposes the UvrA-binding determinant of D2. We propose that the Mfd-EC complex then loops through an ATP hydrolysis cycle (C1 → C2 → C3 → C4 → C5 → C1 →…) that works to overwind the upstream duplex DNA, facilitating RNA transcript release and transcription bubble collapse to displace the RNAP from the nucleic acids. During this cycle, movements of Mfd on the upstream duplex DNA also force conformational changes in the RNAP; clamp opening in C3(adp), and βlobe-Si1 opening in C5(ATP). These RNAP conformational changes also serve to weaken the RNAP-nucleic acid contacts, facilitating RNAP release.

The connection between topological wrapping or clamping of a nucleic acid processing enzyme on the nucleic acid template and high processivity is well established [see, for example, (*Breyer and Matthews, 2001*)]. Not all highly processive nucleic acid processing enzymes topologically wrap or clamp the nucleic acids, but wrapping or clamping the nucleic acids is often associated with 'extremely' high processivity (thousands of nucleotides or more). Examples include replicative DNA polymerases attached to their processivity clamps [PCNA in eukaryotes and archaea, Pol III β-subunit in bacteria, related structures in many bacteriophage; *Bruck and O'Donnell, 2001*] and cellular DNA-dependent RNAPs, which essentially have infinite processivity (*Landick, 2001*).

In its elaborate conformational transitions during the process of EC disruption, Mfd topologically wraps itself around the duplex DNA. Thus, it is natural to propose that the extremely high processivity of Mfd translocation is due to this topological wrapping.

After walking freely in the downstream direction for nine base pairs, the Mfd translocation module butts up against the RNAP βprotrusion (*Figure 4F*). The D1–D3 module accommodates itself to the new molecular environment it finds itself in, finally exposing the previously occluded UvrA-binding determinant (*Figures 5* and *8*).

At this point, if the RNAP is backtracked (*Komissarova and Kashlev, 1997a*; *Nudler, 2012*), continued Mfd translocation in turn forward translocates the RNAP until the active configuration of the EC, with the RNA transcript 3'-end in the RNAP active site, is reached (*Park et al., 2002*). In the presence of NTP substrates, RNA chain elongation by the RNAP can resume and the RNAP, which translocates at ~15–20 nucleotides/s, runs away from Mfd, which translocates at only approximately four nucleotides per second (*Howan et al., 2012*; *Le et al., 2018*).

If RNAP forward translocation is blocked, either by the absence of NTP substrates, a bulky lesion in the t-strand DNA (such as a CPD dimer), or by a roadblock such as a DNA binding protein, continued translocation by the Mfd translocation module will essentially pull and torque the DNA out the back of the RNAP. The torquing action of Mfd positively supercoils the DNA between the Mfd translocation module and the tight grip of the RNAP on the downstream duplex DNA, causing the nt-strand DNA of the transcription bubble to reanneal with the t-strand DNA, displacing the RNA transcript (*Park and Roberts, 2006*). The stability of the EC is dependent on the RNA/DNA hybrid (*Kireeva et al., 2000*), so release of the RNA transcript results in release of the RNAP from the DNA template.

The particles that gave rise to all seven of the Mfd-EC structures came from two separate samples that were prepared in the same way (*Figure 2—figure supplement 1*), so the relative numbers of particles that belong to each structural class are related to the relative stability of that class – the more particles in a structural class, the more stable that structure is. The first structure in our pathway [L1(atp)] is also the least stable (*Figure 4—figure supplement 1A*), indicating that this first step of engagement of Mfd with the DNA is reversible and that L1(atp) is likely in a dynamic equilibrium with [L0] and/or Mfd in solution. The single molecule analysis of *Howan et al., 2012* found that engagement of Mfd with a stalled EC was characterized by weak initial binding of Mfd (Mfd molecules attempt binding to the stalled EC many times before engaging productively), consistent with our findings.

The initial weak binding of Mfd to the stalled EC was followed by a very slow catalytic step ($k = 0.059 \text{ s}^{-1}$) that required multiple rounds of ATP hydrolysis (*Howan et al., 2012*). The L1(atp) → L2(adp) transition requires nine rounds of ATP hydrolysis. The reversible equilibrium at the first step and the presumably slow, multi-step transition required to ultimately reach the NHC are consistent with proposals that Mfd kinetically discriminates a stalled EC (which Mfd is charged with displacing) and an EC that is simply paused (*Kang et al., 2019*; *Landick, 2006*) and should not be displaced. The slow, reversible loading pathway ensures that only the very long-lived stalled EC becomes a target for Mfd displacement and recruitment of UvrA.

The distribution of pause lifetimes in vivo is not known, but the in vivo half-lives of some well-characterized long-lived pauses were estimated (*Larson et al., 2014*): the *his*-pause half-life (known to be among the longest-lived pauses in vivo) was estimated to be ~1.3 s. On the other hand, single-molecule experiments of *Howan et al., 2012*, which were done under saturating ATP concentrations, determined that the mean lifetime of the Mfd-EC displacement complex (our L1 → C5) between Mfd engagement (L1) and RNAP displacement (post-C5) is about 45 s. This is a lower bound for the mean lifetime of a stalled EC due to the high Mfd concentration used in these experiments. Thus, the expected time required for Mfd to find and displace a stalled or paused EC is more than an order of magnitude longer than the longest-lived transcriptional pauses in vivo. The kinetic discrimination of stalled vs. paused (but still productive) ECs explains why overexpression of Mfd is not toxic to cells (*Deaconescu et al., 2006*; *Roberts and Park, 2004*; *Selby and Sancar, 1993*; *Smith et al., 2012*).

The NHC states (C1–C5) are considerably more stable than the loading states (L1 and L2; *Figure 4—figure supplement 1A*), indicating that once the complex passes through the loading pathway and enters the NHC, it is essentially irreversibly trapped, where it attempts to translocate on the DNA against the immovable EC, imparting positive torque on the DNA and ultimately collapsing the transcription bubble, releasing the RNA transcript, and displacing the RNAP from the DNA. During this process, Mfd manhandles the RNAP, prying open the RNAP clamp and βlobe-Si1 to cause RNAP to lose its grip on the nucleic acids (*Figure 7*).

Thus, entry into the NHC is essentially like a molecular switch. In the reversible, slow loading pathway, Mfd probes the lifetime of the stalled/paused EC. During these stages, the Mfd UvrA-binding determinant is occluded; premature recruitment of UvrA would disrupt the Mfd-EC complex, short circuiting the TCR pathway. It could also counterproductively recruit NER components to sites on the genome that are not damaged. The elaborate conformational choreography of Mfd loading and EC displacement is thus evolutionarily tuned to prevent disruption of normal regulation of transcription elongation but to engage the NER pathway when a lesion is likely encountered.

# Materials and methods

## Key resources table

| Reagent type (species) or resource | Designation | Source or reference | Identifiers | Additional information |
|---|---|---|---|---|
| Strain, strain background (*Escherichia coli*) | *Eco* BL21(DE3) | Novagen | | |
| Strain, strain background (*Escherichia coli*) | *Eco* Rosetta(DE3)pLysS | Novagen | | |
| Recombinant DNA reagent | pACYCDuet-1_Ec_rpoZ | PMID:21416542 | | |
| Recombinant DNA reagent | pEcrpoABC(-XH)Z | PMID:21416542 | | |
| Recombinant DNA reagent | pAD6 | PMID:16469698 | | |
| Recombinant DNA reagent | pAR1707 | PMID:3309350 | | |
| Sequence-based reagent | Mfd_scaffold1b_top | This paper | Synthetic DNA oligonucleotide | CCCAACGGCACCGCTGCAA GGAATAGGATACTTGCGGGCTA GGCTCTTATGGCGGCGAATACCC |
| Sequence-based reagent | Mfd_scaffold1b_bot | This paper | Synthetic DNA oligonucleotide | GGGTATTCGCCGCGTACCTCT CCTAGCCCGCAAGTATCCTATT CCTTGCAGCGGTGCCGTTGGG |
| Sequence-based reagent | Mfd_RNA_20A11 | This paper | Synthetic RNA oligonucleotide | GCAUUCAAAGCGGAGAGGUA |
| Chemical compound, drug | 3-[(3-Cholamidopropyl) dimethylammonio]−2-Hydroxy-1-Propanesulfonate (CHAPSO) | Anatrace | Cat# C317 | |
| Chemical compound, drug | 3'-deoxyuridine-5'-triphosphate | Trilink Biotechnologies | Cat# N3005 | |
| Software, algorithm | Bayesian Polishing | PMID:30412051 | | |
| Software, algorithm | Bsoft | PMID:23954653 | | |
| Software, algorithm | Coot | PMID:15572765 | | |
| Software, algorithm | cryoSPARC | PMID:28165473 | | |
| Software, algorithm | Gautomatch | http://www.mrc-lmb.cam.ac.uk/kzhang/Gautomatch | | |
| Software, algorithm | Gctf | PMID:26592709 | | |
| Software, algorithm | Molprobity | PMID:20057044 | | |
| Software, algorithm | MotionCor2 | PMID:28250466 | | |
| Software, algorithm | PHENIX | PMID:20124702 | | |
| Software, algorithm | RELION | PMID:23000701 | | |
| Software, algorithm | PyMOL | http://pymol.org/2/ | | |
| Software, algorithm | SerialEM | PMID:16182563 | | |
| Software, algorithm | UCSF Chimera | PMID:15264254 | | |
| Other | C-flat CF-1.2/1.3 400 mesh gold grids | Electron Microscopy Sciences | Cat# CF413-100-Au | |

## Lead contact and materials' availability

All unique/stable reagents generated in this study are available without restriction from the Lead Contact, Seth A. Darst (darst@rockefeller.edu).

## Experimental model and subject details

RNAP core ($\alpha_2\beta\beta'\omega$) and Mfd are proteins found in *Eco*. For protein expression, *Eco* BL21(DE3) [*Eco str. B F⁻ ompT gal dcm lon hsdS_B(r_B⁻m_B⁻)* λ(DE3 [*lacI lacUV5-T7p07 ind1 sam7 nin5*]) [*malB⁺*]_{K-12}(λ^S)] was used.

## Method details

Structural biology software was accessed through the SBGrid consortium (*Morin et al., 2013*).

## Protein expression and purification

Eco RNAP (harboring full-length α-subunits) was purified as described previously (*Twist et al., 2011*; *Chen et al., 2019a*). A pET-based plasmid overexpressing each subunit of *Eco* RNAP (full-length α, β, ω) as well as β'-PPX-His10 (PPX; PreScission protease site, LEVLFQGP, GE Healthcare Life Sciences) was co-transformed with a pACYCDuet-1 plasmid containing *Eco* rpoZ (encoding ω) into *Eco* BL21(DE3) (Novagen). Protein expression was induced with 1 mM isopropyl ß-D-thiogalactopyranoside (IPTG) for 4 hr at 30°C. Cells were harvested and lysed with a French Press (Avestin) at 4°C. Lysate was precipitated using polyethyleneimine [PEI, 10% (w/v), pH 8.0, Acros Organics]. Pellets were washed and RNAP was eluted. The PEI elutions were precipitated with ammonium sulfate. Pellets were harvested, resuspended, and loaded onto HiTrap IMAC HP columns (GE Healthcare Life Sciences) for purification by $Ni^{2+}$-affinity chromatography. Bound RNAP was washed on column, eluted, and dialyzed. Dialyzed RNAP was loaded onto a Biorex-70 column (Bio-Rad) for purification by ion exchange chromatography. Eluted RNAP was concentrated by centrifugal filtration, then loaded onto a HiLoad 26/600 Superdex 200 column (GE Healthcare Life Sciences) for purification by size exclusion chromatography. Purified RNAP was supplemented with glycerol to 20% (v/v), flash frozen in liquid $N_2$, and stored at −80°C.

Eco Mfd was purified as described previously (*Deaconescu and Darst, 2005*). A pET-based plasmid overexpressing N-terminal His_6-tagged *Eco* Mfd was transformed into Rosetta(DE3)pLysS cells (Novagen). Protein expression was induced with 1 mM isopropyl β-D-1-thiogalactopyranoside (IPTG) for 4 hr at 30°C, and the cells were harvested and lysed in a buffer containing 50 mM Tris, pH 8.0, 500 mM NaCl, 15 mM imidazole, 10% (v/v) glycerol, 2 mM β-mercaptoethanol (β-ME), 1 mM PMSF, and protease inhibitor cocktail (cOmplete, EDTA-free protease inhibitor cocktail, Roche). For purification, the clarified lysate was loaded onto a $Ni^{2+}$-charged Hitrap IMAC (GE Healthcare Life Sciences) column and the protein was eluted using 0–200 mM imidazole gradient. The elutions were dialyzed in a buffer containing 20 mM Tris, pH 8.0, 100 mM NaCl, 10% (v/v) glycerol, 5 mM EDTA, and 10 mM β-ME, and loaded onto a Hitrap Heparin (GE Healthcare Life Sciences) column. The protein was eluted with a 100 mM – 2 M NaCl gradient, and further purified by size exclusion chromatography on a HiLoad 16/600 Superdex200 (GE Healthcare Life Sciences) column in a buffer containing 20 mM Tris, pH 8.0, 500 mM NaCl, and 10 mM DTT. Purified Mfd was supplemented with glycerol to 20% (v/v), flash frozen in liquid $N_2$, and stored at −80°C.

## Assembly of ECs and Mfd-EC complexes

To assemble ECs, the RNA and the t-strand DNA (*Figure 1A*) were annealed at a 1:1 molar ratio in a thermocycler (95°C for 2 min, 75°C for 2 min, 45°C for 5 min, followed by cooling to 25°C at 1 °C/min). Purified *Eco* RNAP was buffer exchanged into transcription buffer (20 mM Tris, pH 8.0, 150 mM KCl, 10 mM MgCl_2, 5 mM DTT). The annealed DNA–RNA hybrid was added to the buffer-exchanged *Eco* RNAP at a 1.2:1 molar ratio and incubated for 15 min at room temperature. Then the nt-strand DNA was added and incubated for 10 min. To assemble Mfd-EC complexes, 2 mM ATP was added to the EC, and purified Mfd was added at a 1:2 ratio.

## Biochemical analysis of Mfd/EC complexes

EC-Mfd complexes were visualized by EMSA. ECs were incubated with twofold molar excess of Mfd and 2 mM ATP at room temperature for 5 min in transcription buffer. Samples were diluted with 6× loading dye containing 20 mM Tris, pH 8.0, xylene cyanol, bromophenol blue, and 30% (v/v) glycerol and loaded onto 4.5% (29:1 acrylamide:bis-acrylamide) polyacrylamide native gels at 4°C in 1× TBE (89 smM Tris, 89 mM boric acid, 2 mM EDTA).

To prepare $VO_4^{3-}$, sodium orthovanadate (Sigma-Aldrich) was dissolved in water, and the pH was adjusted to 10 using HCl, resulting in a dark orange color. The solution was boiled for 2 min until it turned colorless, and then cooled to room temperature. The process of adjusting the pH and boiling was repeated two more times. The $VO_4^{3-}$ concentration was determined spectroscopically using its molar extinction coefficient of 2,925 $M^{-1}$ $cm^{-1}$. The solution was stored frozen at −80°C until further use. Reactions with ATP-$VO_4^{3-}$ contained 8 mM sodium orthovanadate (pH 10). Alternatively, a 10 mM mixture of ADP-$VO_4^{3-}$ was added to the ECs and Mfd for a final concentration of 2 mM.

For in vitro EC displacement assays (*Figure 6B*), we followed the procedures of *Chambers et al., 2003* with some modifications. Stalled ECs were formed by nucleotide starvation on a 529 base pair RasI-SmaI fragment of plasmid pAR1707 (*Levin et al., 1987*) end-labeled using T4 polynucleotide kinase and [γ-$^{32}$P]ATP. ECs stalled at +20 were formed by initiating with the dinucleotide ApU and ATP, CTP, and GTP (excluding UTP), and with the addition of the chain-terminator 3'-deoxy-UTP. Complexes were analyzed by EMSAs using 4.5% polyacrylamide (37.5:1 acrylamide/bisacrylamide) in 1× TBE at 4°C. Radiolabeled bands were detected using a phosphor screen and quantified using Imagequant software (Molecular Dynamics). *Eco* RNAP holoenzyme was assembled by incubating *Eco* RNAP (250 nM final) with σ$^{70}$ (1.25 μM final) at 37°C for 20 min. The $^{32}$P-labeled dsDNA linear scaffold (0.4 nM final) was combined with holoenzyme (10 nM final) at 37°C for 10 min before ApU dinucleotide (200 μM final), ATP (2 mM final), CTP (50 μM final), GTP (50 μM final), and 3'-deoxy-UTP (100 μM final). Heparin (Sigma-Aldrich) was also added (10 μg/mL final) to ensure single-round transcription. After 10 min of incubation at 37°C, Mfd (250 nM final) was added to activate stalled RNAP displacement and translocation. Samples were collected at different timepoints, combined with excess EDTA to stop ATPase activity, and placed on ice before gel loading. The assays were conducted without and with the presence of $VO_4^{3-}$ (20 mM final).

## nMS analysis

For the EC + Mfd sample, 7 μM EC was incubated with 3.5 μM Mfd (2:1 molar ratio) in transcription buffer with 2 mM ATP for 1 min at RT. The nonvolatile buffer components, including ATP, needed to be removed prior to nMS analysis because these components can form nonspecific adductions on protein complexes and degrade spectral quality. The EC and EC + Mfd samples were then buffer-exchanged into nMS solution (500 mM ammonium acetate, pH 7.5, 0.01% Tween-20) using Zeba microspin desalting columns (Thermo Fisher Scientific) with a 40 kDa MWCO (*Olinares et al., 2016*).

For nMS analysis, 2–3 μL of the buffer-exchanged sample was loaded into a gold-coated quartz emitter that was prepared in-house and then electrosprayed into an Exactive Plus EMR instrument (Thermo Fisher Scientific) with a static nanospray source (*Olinares and Chait, 2019*). The MS parameters used include: spray voltage, 1.2–1.3 kV; capillary temperature, 125°C; in-source dissociation, 10 V; S-lens RF level, 200; resolving power, 8750 at *m/z* of 200; AGC target, 1 × 10$^6$; maximum injection time, 200 ms; number of microscans, 5; injection flatapole, 8 V; interflatapole, 4 V; bent flatapole, 4 V; high energy collision dissociation (HCD), 180–200 V; ultrahigh vacuum pressure, 6–7 × 10$^{-10}$ mbar; total number of scans, at least 100. Mass calibration in positive EMR mode was performed using cesium iodide. For data processing, the acquired MS spectra were visualized using Thermo Xcalibur Qual Browser (versions 3.0.63 and 4.2.47). MS spectra deconvolution was performed either manually or using the software UniDec versions 3.2 and 4.1 (*Marty et al., 2015*; *Reid et al., 2019*). The deconvolved spectra obtained from UniDec were plotted using the m/z software (Proteometrics LLC). Experimental masses were reported as the average mass ± standard deviation (S.D.) across all the calculated mass values within the observed charge state series. Mass accuracies were calculated as the percent difference between the measured and expected masses relative to the expected mass. The measured masses for the corresponding complexes (with mass accuracies reported in parentheses) include EC: 437,680 ± 20 Da (0.016%) for the EC only sample and EC + 1 Mfd: 570,360 ± 20 Da (0.03%), EC: 437,700 ± 20 Da (0.02%), α$_2$β subcomplex: 223,700 ± 3 Da (0.02%), and Mfd: 132,582 ± 8 Da (0.003%) for the EC sample incubated with Mg-ATP and Mfd.

## Preparation of Mfd-EC Cryo-EM grids

For cryo-EM grid preparation, ECs prepared as above were purified over a Superose 6 INCREASE gel filtration column (GE Healthcare Life Sciences) equilibrated with transcription buffer. ATP (2 mM)

and twofold excess Mfd were added and incubated for 1 min before adding {3-([3-cholamidopropyl] dimethylammonio)−2-hydroxy-1-propanesulfonate} (CHAPSO; Anatrace) to a final concentration of 8 mM (*Chen et al., 2019b*). The final buffer condition for all the cryo-EM samples was the same as transcription buffer. C-flat holey carbon grids (CF-1.2/1.3-4Au, Protochips) were glow-discharged for 20 s prior to the application of 3.5 µL of the samples. Using a Vitrobot Mark IV (Thermo Fisher Scientific Electron Microscopy), grids were blotted and plunge-froze into liquid ethane with 100% chamber humidity at 22°C.

## Cryo-EM data acquisition and processing

For the cryo-EM structure determination of the Mfd-EC, two data sets were collected and combined (Figure S1). In each data collection, grids were imaged using a 300 keV Titan Krios (Thermo Fisher Scientific Electron Microscopy) equipped with a K2 Summit direct electron detector (Gatan, Pleasanton, CA). Images were recorded with Serial EM (*Mastronarde, 2005*) with a pixel size of 1.3 Å over a defocus range of −0.8 to −2.4 µm. Movies were recorded in super-resolution mode at eight electrons/physical pixel/s in dose-fractionation mode with subframes of 0.2 s over a 10 s exposure (50 frames) to give a total dose of 80 electrons/physical pixel or 47.3 electrons/Å$^2$. Dose-fractionated movies were gain-normalized, drift-corrected, binned, summed, and dose-weighted using MotionCor2 (*Zheng et al., 2017*). The contrast transfer function was estimated for each summed image using Gctf (*Zhang, 2016*). Gautomatch (developed by K. Zhang, MRC Laboratory of Molecular Biology, Cambridge, UK, http://www.mrc-lmb.cam.ac.uk/kzhang/Gautomatch) was used to pick particles without templates. Picked particles from each data set were extracted from the dose-weighted images in RELION (*Scheres, 2012*) using a box size of 300 pixels and subjected to 2D classification in RELION to exclude false particles. The selected particles from 2D classification were transferred to CryoSPARC (*Punjani et al., 2017*) to perform heterogeneous refinement with multiple 3D references to separate Mfd-EC complexes from isolated EC and Mfd particles. For the heterogeneous refinement, cryo-EM maps of *Eco* EC [EMD-8585; (*Kang et al., 2017*)], Mfd-EC, and isolated Mfd (the latter two generated from a subset of the data by *ab initio* reconstruction) were used as templates (each low-pass filtered to 30 Å resolution). The class representing Mfd-EC was further refined in CryoSPARC homogeneous refinement, yielding 3.9 Å and 3.6 Å nominal resolution maps from the first and second data sets, respectively. The refined particles from the first data set were transferred to RELION for 3D auto-refinement, CTF refinement, Bayesian polishing, and 3D autorefinement before being combined with the particles from the second data set. The resulting 594,435 particles were 3D autorefined in RELION for a consensus refinement, yielding a 3.86 Å nominal resolution map. A second round of CTF refinement, Bayesian polishing, and 3D autorefinement improved the resolution to 3.64 Å. By using focused classification around the Mfd region, eight classes were generated with distinct Mfd conformations. Among the eight classes, two classes were discarded because they could not be refined further. Four of the classes were further refined using RELION 3D autorefinement to yield C1, C3, C4, and C5 (*Figure 2—figure supplement 1*). The remaining two classes were further sorted using partial signal subtraction (*Bai et al., 2015*) of the region outside of Mfd and the RNAP βprotrusion, then classifying these subtracted particles using a mask encompassing Mfd and the RNAP β-protrusion density, resulting in L1, L2, and C2 (*Figure 2—figure supplement 1*). In total, the seven resulting maps showed well-defined EC density but variable quality maps for the Mfd component, indicating that the alignments were dominated by the EC portion of the complex. Therefore, to better resolve the density around Mfd, particles from each class were imported into cryoSPARC and refined using cryoSPARC Non-uniform Refinement (*Punjani et al., 2019*). Using the resulting maps, masks around Mfd were generated for cryoSPARC Local Refinement. The fulcrum points (alignment centers) for each of these masks were defined using 'Volume Tracer' in UCSF Chimera (*Pettersen et al., 2004*). Each class from the Non-uniform Refinement was subjected to Local Refinement using their respective Mfd mask and fulcrum point. The local refinements resulted in better resolved Mfd density for each class with the following nominal resolutions in this region: L1 (atp, 6.6 Å), L2 (adp, 6.7 Å), C1 (ATP, 5.2 Å), C2 (ADP, 3.5 Å), C3 (adp, 4.4 Å), C4 (ADP, 3.8 Å), and C5 (ADP, 3.3 Å).

The EC-centered cryo-EM maps from RELION and the Mfd-centered maps from cryoSPARC were combined using the PHENIX combine_focused_maps command (*Adams et al., 2010*). The procedure that gave the highest quality maps was as follows: coordinates were rigid body refined into each map. For the combination step: (1) for the EC-centered coordinates, the occupancies of the

EC-RNAP, the EC nucleic acids, and Mfd-D4(RID) were set to 1.0, while the occupancies for the rest of Mfd and the upstream duplex DNA were set to 0; (2) for the Mfd-centered coordinates, the occupancies of the entire Mfd, the RNAP βprotrusion, and the upstream duplex DNA were set to 1.0, while the rest of the EC was set to occupancy = 0. Thus, in the combined maps, the EC density and EC nucleic acids came from the EC-centered cryo-EM maps, while Mfd and the upstream duplex DNA density came from the Mfd-centered maps, and the density for the Mfd-D4(RID) and the RNAP β-protrusion were weighted according to the combine_focused_maps algorithm. These combined maps were the most interpretable over the entirety of each complex and were therefore used for building, refinement, statistics reporting (*Supplementary file 1s*), and deposition in the Electron Microscopy Data Bank (EMDB). RELION 3D auto-refinement and post-processing of the polished particles resulted in structures with the following nominal resolutions: L1 (atp, 4.1 Å), L2 (adp, 4.0 Å), C1 (ATP, 3.9 Å), C2 (ADP, 3.9 Å), C3 (adp, 3.2 Å), C4 (ADP, 3.6 Å), and C5 (ADP, 3.3 Å). Local resolution calculations were done using blocres and blocfilt from the Bsoft package (*Cardone et al., 2013*).

### Model building and refinement

For initial models of the complexes, the *Eco* EC structure (PDB ID 6ALF; *Kang et al., 2017*) was manually fit into the combined cryo-EM density maps using Chimera (*Pettersen et al., 2004*) and real-space refined using Phenix (*Adams et al., 2010*). The DNAs and Mfd domains [taken from 2EYQ (*Deaconescu et al., 2006*)] were mostly built de novo based on the density maps. For real-space refinement, rigid body refinement was followed by all-atom and B-factor refinement with Ramachandran and secondary structure restraints. Refined models were inspected and modified in Coot (*Emsley and Cowtan, 2004*).

### Superimposition of structures and calculation of rmsds

For the statistics presented in *Supplementary files 3–5*, α-carbons of the specified structural components were superimposed using the PyMOL align command (the resulting rmsd is listed in the 'align' column, with the number of α-carbon atoms used for the rmsd calculation listed underneath in parentheses). The rmsd for all of the α-carbon atoms was then determined using the PyMOL 'rms_cur' command (listed under the 'rms_cur' column).

### Quantification and statistical analysis

The nMS spectra were visualized using Thermo Xcalibur Qual Browser (versions 3.0.63 and 4.2.27), deconvolved using UniDec versions 3.2 and 4.1 (*Marty et al., 2015*; *Reid et al., 2019*) and plotted using the m/z software (Proteometrics LLC, New York, NY). Experimental masses (*Figure 1D*) were reported as the average mass ± standard deviation across all the calculated mass values obtained within the observed charge state distribution.

ImageQuant 5.2 (GE Healthcare, Pittsburgh PA) was used to visualize and quantify gels. To quantify the EC displacement assays (*Figure 6B*), mean values and the standard error of the mean from three independent measurements were calculated.

Structural biology software was accessed through the SBGrid consortium (*Morin et al., 2013*). The local resolution of the cryo-EM maps (Figure S2) was estimated using blocres (*Cardone et al., 2013*) with the following parameters: box size 15, verbose 7, sampling 1.3, and cutoff 0.5. The quantification and statistical analyses for model refinement and validation were generated using MolProbity (*Chen et al., 2010*) and PHENIX (*Adams et al., 2010*).

## Acknowledgements

We thank M Oldham and AJ Smith for assistance and advice with experimental procedures, M Ebrahim and J Sotiris at The Rockefeller University Evelyn Gruss Lipper Cryo-electron Microscopy Resource Center for help with cryo-EM data collection, and N Savery, T Strick, and members of the Darst/Campbell laboratory for helpful discussion on the manuscript. This work was supported by NIH grants P41 GM109824 and P41 GM103314 to BTC, R01 GM114450 to EAC, and R35 GM118130 to SAD.

# Additional information

## Funding

| Funder | Grant reference number | Author |
|---|---|---|
| National Institute of General Medical Sciences | P41 GM109824 | Brian T Chait |
| National Institute of General Medical Sciences | P41 GM103314 | Brian T Chait |
| National Institute of General Medical Sciences | R01 GM114450 | Elizabeth A Campbell |
| National Institute of General Medical Sciences | R35 GM118130 | Seth A Darst |

The funders had no role in study design, data collection and interpretation, or the decision to submit the work for publication.

## Author contributions

Jin Young Kang, Eliza Llewellyn, Paul Dominic B Olinares, Investigation, Methodology, Writing - review and editing; James Chen, Investigation, Methodology; Joshua Brewer, Investigation; Brian T Chait, Elizabeth A Campbell, Supervision, Funding acquisition, Methodology, Writing - review and editing; Seth A Darst, Conceptualization, Supervision, Funding acquisition, Investigation, Methodology, Writing - original draft, Writing - review and editing

## Author ORCIDs

Paul Dominic B Olinares ⓘ https://orcid.org/0000-0002-3429-6618
Seth A Darst ⓘ https://orcid.org/0000-0002-8241-3153

## Decision letter and Author response

Decision letter https://doi.org/10.7554/eLife.62117.sa1
Author response https://doi.org/10.7554/eLife.62117.sa2

# Additional files

## Supplementary files

• Supplementary file 1. Cryo-EM data collection, refinement, and validation statistics.

• Supplementary file 2. Bound nucleotide test.

• Supplementary file 3. Conformational changes for the RNAP component of the Mfd-EC structures.

• Supplementary file 4. Conformational changes for the Mfd component of the Mfd-EC structures.

• Supplementary file 5. Conformational changes for the entire Mfd-EC complexes.

• Supplementary file 6. Conformational transitions (translation of center-of-gravity and rotation) for Mfd domains.

• Transparent reporting form

## Data availability

The cryo-EM density maps have been deposited in the EMDataBank under accession codes EMD-21996 [L1(ATP)], EMD-22006 [L2(ADP)], EMD-22012 [C1(ATP)], EMD-22039 [C2(ATP)], EMD-22043 [C3(ADP)], EMD-22044 [C4(ADP)], and EMD-22045 [C5(ATP)]. The atomic coordinates have been deposited in the Protein Data Bank under accession codes 6X26 [L1(ATP)], 6X2F [L2(ADP)], 6X2N [C1(ATP)], 6X43 [C2(ATP)], 6X4W [C3(ADP)], 6XYY [C4(ADP)], and 6X50 [C5(ATP)].

The following datasets were generated:

| Author(s) | Year | Dataset title | Dataset URL | Database and Identifier |
|---|---|---|---|---|
| Kang JY, Llewellyn E, Chen J, Darst SA | 2020 | Mfd-bound E.coli RNA polymerase elongation complex - L1 state | https://www.rcsb.org/structure/6X26 | RCSB Protein Data Bank, 6X26 |
| Kang JY, Llewellyn E, Chen J, Darst SA | 2020 | Mfd-bound E.coli RNA polymerase elongation complex - L1 state | https://www.ebi.ac.uk/pdbe/entry/emdb/EMD-21996 | Electron Microscopy Data Bank, EMD-21996 |
| Kang JY, Llewellyn E, Chen J, Darst SA | 2020 | Mfd-bound E.coli RNA polymerase elongation complex - L2 state | https://www.rcsb.org/structure/6X2F | RCSB Protein Data Bank, 6X2F |
| Kang JY, Llewellyn E, Chen J, Darst SA | 2020 | Mfd-bound E.coli RNA polymerase elongation complex - L2 state | https://www.ebi.ac.uk/pdbe/entry/emdb/EMD-22006 | Electron Microscopy Data Bank, EMD-22006 |
| Kang JY, Llewellyn E, Chen J, Darst SA | 2020 | Mfd-bound E.coli RNA polymerase elongation complex - C1 state | https://www.rcsb.org/structure/6X2N | RCSB Protein Data Bank, 6X2N |
| Kang JY, Llewellyn E, Chen J, Darst SA | 2020 | Mfd-bound E.coli RNA polymerase elongation complex - C1 state | https://www.ebi.ac.uk/pdbe/entry/emdb/EMD-22012 | Electron Microscopy Data Bank, EMD-22012 |
| Kang JY, Llewellyn E, Chen J, Darst SA | 2020 | Mfd-bound E.coli RNA polymerase elongation complex - C2 state | https://www.rcsb.org/structure/6X43 | RCSB Protein Data Bank, 6X43 |
| Kang JY, Llewellyn E, Chen J, Darst SA | 2020 | Mfd-bound E.coli RNA polymerase elongation complex - C2 state | https://www.ebi.ac.uk/pdbe/entry/emdb/EMD-22039 | Electron Microscopy Data Bank, EMD-22039 |
| Kang JY, Llewellyn E, Chen J, Darst SA | 2020 | Mfd-bound E.coli RNA polymerase elongation complex - C3 state | https://www.rcsb.org/structure/6X4W | RCSB Protein Data Bank, 6X4W |
| Kang JY, Llewellyn E, Chen J, Darst SA | 2020 | Mfd-bound E.coli RNA polymerase elongation complex - C3 state | https://www.ebi.ac.uk/pdbe/entry/emdb/EMD-22043 | Electron Microscopy Data Bank, EMD-22043 |
| Kang JY, Llewellyn E, Chen J, Darst SA | 2020 | Mfd-bound E.coli RNA polymerase elongation complex - C4 state | https://www.rcsb.org/structure/6XYY | RCSB Protein Data Bank, 6XYY |
| Kang JY, Llewellyn E, Chen J, Darst SA | 2020 | Mfd-bound E.coli RNA polymerase elongation complex - C4 state | https://www.ebi.ac.uk/pdbe/entry/emdb/EMD-22044 | Electron Microscopy Data Bank, EMD-22044 |
| Kang JY, Llewellyn E, Chen J, Darst SA | 2020 | Mfd-bound E.coli RNA polymerase elongation complex - C5 state | https://www.rcsb.org/structure/6X5Q | RCSB Protein Data Bank, 6X5Q |
| Kang JY, Llewellyn E, Chen J, Darst SA | 2020 | Mfd-bound E.coli aMfd-bound E. coli RNA polymerase elongation complex - C5 stateRNA polymerase elongation complex - L1 state | https://www.ebi.ac.uk/pdbe/entry/emdb/EMD-22045 | Electron Microscopy Data Bank, EMD-22045 |

The following previously published datasets were used:

| Author(s) | Year | Dataset title | Dataset URL | Database and Identifier |
|---|---|---|---|---|
| Deaconescu AM, Darst SA | 2006 | Crystal structure of Escherichia coli transcription-repair coupling factor | https://www.rcsb.org/structure/2EYQ | RCSB Protein Data Bank, 2EYQ |
| Kang JY, Darst SA | 2017 | CryoEM structure of crosslinked E. coli RNA polymerase elongation complex | https://www.rcsb.org/structure/6ALF | RCSB Protein Data Bank, 6ALF |
| Deaconescu AM, Grigorieff N | 2012 | Core UvrA/TRCF complex | https://www.rcsb.org/structure/4DFC | RCSB Protein Data Bank, 4DFC |
| Kang JY, Darst SA | 2017 | CryoEM structure of crosslinked E. coli RNA polymerase elongation complex | https://www.ebi.ac.uk/pdbe/entry/emdb/EMD-8585/experiment | Electron Microscopy Data Bank, EMD-8585 |

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
