## [Decision Letter]

**Acceptance summary:**

The authors pursued the analysis of a stalled RNA polymerase followed by its release through the transcription repair coupling factor Mfd by cryo electron microscopy. The authors succeeded to obtain several snapshots of this reaction and thereby derived an overall model of this process which addresses many open questions such as the prevention of a premature engagement of the repair machinery or how the release of the polymerase is facilitated. This will be a landmark paper in understanding the molecular mechanisms and interactions that underlie the activity of the Mfd protein, and also thereby basic activities of the universal transcription machine.

**Decision letter after peer review:**

Thank you for submitting your article "Structural basis for transcription complex disruption by the Mfd translocase" for consideration by *eLife*. Your article has been reviewed by two peer reviewers, and the evaluation has been overseen by a Reviewing Editor and Cynthia Wolberger as the Senior Editor. The reviewers have opted to remain anonymous.

The reviewers have discussed the reviews with one another and the Reviewing Editor has drafted this decision to help you prepare a revised submission.

Summary:

The present paper describes cryo EM snapshots of seven different states of EC complexes obtained with a specially designed DNA/RNA substrate, RNA polymerase (RNAP) and the Mfd translocase in the presence of ATP. Using conformational differences in these snapshots, they are grouped and ordered, and analyzed for their potential role during ATP hydrolysis Mfd. Elaborate models are proposed for how the different states form an Mfd activity cycle that results in the extraction of DNA from the back of RNAP while simultaneously positively supercoiling the DNA.

The paper is a remarkable achievement that reveals the detailed molecular activity of Mfd in dissociating a stalled RNA polymerase. It is the climax of decades of experimentation on Mfd, explaining and summarizing studies of many sorts. It is also a striking demonstration of the power of high resolution cryo-EM to reveal enzymatic mechanism, including distinct stages of Mfd translocation and RNAP dissociation; the timing of the interaction of UvrA with Mfd to initiate lesion excision relative to Mfd engagement with RNA polymerase also is satisfyingly explained. The authors reveal how the kinetics of Mfd engagement protects a slow or paused enzyme from even artificially high cellular concentrations of Mfd through a slow step of Mfd action to which only a blocked RNAP will be susceptible. Pending a few suggested revisions, it certainly merits publication in *eLife*.

Essential revisions:

1) The data are grouped into ATP or ADP bound states that are used for clustering. For the L1 state it is proposed the nucleotide cannot be identified by the experimental data and then it is inferred that since its conformation is the closest to the ATP bound states, it should contain ATP. Looking at the densities in the provided files (there is no L1 density in this area provided in Figure 2), it is difficult to identify anything bound at this position, suggesting possible overinterpretation of the data. In the final model of this snapshot ,the ATP should most likely be removed and all statements concerning this state should be stated more cautiously.

2) The L0 state is clearly marked as “model” but the L1.5a b states also show modeled intermediates to explain transitions. This may be misleading, since there is no evidence presented that this is actually occurring. The authors then arrive at the L2 state that contains ADP, which is partly supported by the cryo EM density. However, compared to C3, which also has a resolution of 4.0 Å, the map quality in that region is of low quality and the interpretation of L2 is highly ambiguous. It would be helpful to assess the real space coefficients for the nucleotides and the area surrounding them with a radius of around 10-15 Å. This would help the reader to make an informed decision on the quality of the experimental data.

3) The L2 state leads to an ATPase cycle state that is reflected by C1-C5. For the comparison of the nucleotide cycle, the TD1 domains of all structures are superimposed and compared to the TD2 movement, with a special focus on residues G874, R902, and R905. It is stated that in all ATP-containing structures, these residues form polar contacts with the γ phosphate of ATP. However, the side chains of the arginine residues are only visible in the cryo EM densities in C2 and C5, and not in C1 or L1 according to the structural data provided (in the ADP-containing complexes, they are only visible in C4). Figure 3A therefore compares C5 and C4, however, the statement was made more general than the data would suggest. In addition, the polar contacts shown in Figure 3A for C5 should be revised. R902 has a distance of more than 4 Å to the γ phosphate, so the dashed line is misleading. In Figure 3B the movements of the TD2 domain are shown based on nucleotide state. Here, it is unclear of which structures the rotation and center of mass shifts were derived. Was it from C4 and C5, or an average of all structures? Is this the average or maximum displacement? Please address these issues.

4) In the next step the TRG motif is compared to an example of residues R929, R953, and Q963. Here again, dashed lines in the figure exceeding a distance of 4 Å are misleading. A minor issue is the side chain orientation of Q963. The hydrogen bonding requirements are much better satisfied when the side chain is flipped; i.e., OE1 would interact with NZ of K739 instead of with the phosphate backbone and NE2 with the phosphate backbone. Although it is appreciated that there are a lot of residues to refine in total, groups that are used to exemplify certain findings should be modeled as accurately as possible.

5) Regarding the statement: "As a result of the large conformational rearrangement of Mfd, Mfd(ADP)L2 is topologically “wrapped” around the DNA (Figure 5), likely explaining how Mfd (with RNAP in tow) translocates processively over many kilobases of DNA (Fan et al., 2016; Graves et al., 2015)." This claim is made without any further explanation or additional data supporting. Please clarify or remove.

6) If the discrimination between arrested and paused transcription complexes is solely kinetic, would it be true that an exceptionally long pause would enable Mfd dissociation? Please explain.

7) The subsection “Mfd loading requires multiple rounds of ATP hydrolysis and is accompanied by profound conformational changes” describes the translocation module TD1-TD2 "bumping" into RNA polymerase in the L2 structure. This presumably is different from the contact between the RID and the RNAP protrusion. Where does it make this contact, and is there a molecular definition of the bump? Is this related to the changes in the RNA polymerase clamp opening during the Mfd ATP cycle? Also, would the clamp opening cycle be different for paused versus blocked complexes, and could this contribute to the relative insensitivity of paused complexes to release by Mfd? Please clarify.

8) Le et al., 2018, showed by single molecule analysis that Mfd can translocate by itself without engagement with RNA polymerase, and proposed a model in which Mfd patrols DNA, moving slower than transcription so that only stopped, or perhaps extensively paused, complexes can be captured. This does appear to be a distinct view of the cellular process. If the model of Le et al. describes the cellular process, some stochastic or other process presumably would have the break the inhibitory structure of the holo Mfd, and this would have to occur at a significant rate. Arguably, Le et al. could have been observing a small fraction of Mfd that is spontaneously active, and therefore not an important contributor to the cellular mechanism. Do the authors have any further thoughts about the two distinct models?

---

## [Author Response]

Essential revisions:1) The data are grouped into ATP or ADP bound states that are used for clustering. For the L1 state it is proposed the nucleotide cannot be identified by the experimental data and then it is inferred that since its conformation is the closest to the ATP bound states, it should contain ATP. Looking at the densities in the provided files (there is no L1 density in this area provided in Figure 2), it is difficult to identify anything bound at this position, suggesting possible overinterpretation of the data. In the final model of this snapshot ,the ATP should most likely be removed and all statements concerning this state should be stated more cautiously.

This is correct that it is not possible to determine the identity of the bound nucleotide, or if there even is a bound nucleotide, from the L1 cryo-EM density alone. We inferred the presence of ATP based primarily on the following arguments:

i) Previous studies have established that Mfd does not stably interact with DNA in the absence of ATP (or a non-hydrolyzable ATP analog) (Selby and Sancar, 1995a, 1995b; Chambers et al., 2003; Smith and Savery, 2007; Howan et al., 2012).

ii) The conformation of the translocation module in L1 is closest to the other ATP-bound states.

While we maintain that these observations strongly support that the L1 state harbors ATP, we have:

i) removed ATP from the deposited model (due to the poor cryo-EM density).

ii) explained our reasoning more clearly (subsection “One ATP hydrolysis cycle corresponds to translocation by one base pair”).

iii) we also performed the objective test to assess if the cryo-EM density can be used to determine the state of the bound nucleotide (ATP or ADP; see below).

2) The L0 state is clearly marked as “model” but the L1.5a b states also show modeled intermediates to explain transitions. This may be misleading, since there is no evidence presented that this is actually occurring. The authors then arrive at the L2 state that contains ADP, which is partly supported by the cryo EM density. However, compared to C3, which also has a resolution of 4.0 Å, the map quality in that region is of low quality and the interpretation of L2 is highly ambiguous. It would be helpful to assess the real space coefficients for the nucleotides and the area surrounding them with a radius of around 10-15 Å. This would help the reader to make an informed decision on the quality of the experimental data.

The L1.5a and L1.5b states were described as “models”, but we have further clarified this by labeling them as “models” in Figure 5.

At the suggestion of the review, we performed a detailed analysis of the nucleotide binding regions in each Mfd structure (except for L1, which is hopeless) as follows:

i) We generated models of each Mfd structural state that were identical in every way except that one model contained ATP and another contained ADP (aligned by the nucleotide common atoms).

ii) Each model was refined (PHENIX real_space_refine) using identical procedures/parameters.

iii) We then used the validation tools in PHENIX to calculate the average real space correlation coefficient for the nucleotide, the Mg^2+^-ion (if modeled), and the following Mfd residues that contact the nucleotide in at least one of the structures: F597, F599, E600, T602, Q605, D629, G631, F632, G633, K634, T635, E636, H665, D729, E730, P780, R783, G874, R902, R905.

iv) We also calculated a sort of “nucleotide R-factor” as follows:

R(%) = 100 X {(cc_a_ – cc_b_)/[(cc_a_ + cc_b_)/2]}

where cc_a_ is the average real space correlation coefficient for the presumed nucleotide in each structure, and cc_b_ is the average real space correlation coefficient for the alternative (test) nucleotide.

The test and the results are described in the subsection “Seven structures in the Mfd activity cycle”, tabulated in the new Supplementary file 2.

We interpret these results to mean that the original assignment of the nucleotide states for C1(ATP), C2(ATP), C4(ADP), and C5(ATP) were correct (R >> 0) but that the assignment of the nucleotide state for the L2 and C3 structures based on the cryo-EM density alone is ambiguous (R close to 0). Note these are the two states with the worst overall nominal resolution (4 Å) other than L1. Again, based on the correspondence of the L2 and C3 translocation module conformation with the other ADP-bound structure [C4(ADP)], we infer that L2 and C3 are also bound to ADP, but for the three ambiguous structures (L1, L2, and C3), they are now referred throughout the manuscript (and figures) with lower case letters for the nucleotide (rather than upper case letters), as explained in the subsection “Seven structures in the Mfd activity cycle”. Also see Figure 2 legend.

We thank the reviewers for suggesting this test, which we believe provides important information for readers to evaluate our claims.

3) The L2 state leads to an ATPase cycle state that is reflected by C1-C5. For the comparison of the nucleotide cycle, the TD1 domains of all structures are superimposed and compared to the TD2 movement, with a special focus on residues G874, R902, and R905. It is stated that in all ATP-containing structures, these residues form polar contacts with the γ phosphate of ATP. However, the side chains of the arginine residues are only visible in the cryo EM densities in C2 and C5, and not in C1 or L1 according to the structural data provided (in the ADP-containing complexes, they are only visible in C4). Figure 3A therefore compares C5 and C4, however, the statement was made more general than the data would suggest. In addition, the polar contacts shown in Figure 3A for C5 should be revised. R902 has a distance of more than 4 Å to the γ phosphate, so the dashed line is misleading. In Figure 3B the movements of the TD2 domain are shown based on nucleotide state. Here, it is unclear of which structures the rotation and center of mass shifts were derived. Was it from C4 and C5, or an average of all structures? Is this the average or maximum displacement? Please address these issues.

As noted above, the noted side-chains interacting with the bound nucleotides (and in the TRG motif, see below) are clearly defined in the C2(ATP), C4(ADP) and C5(ATP) states, but not in the other states with lower resolution cryo-EM densities. We have clarified in the text that these conclusions are derived from the C2(ATP), C4(ADP), and C5(ATP) states, and the conclusions are presumed to apply to the other states (although this is not directly observed in the cryo-EM density – see the subsection “One ATP hydrolysis cycle corresponds to translocation by one base pair”).

We do not agree that the notation of polar contacts (with distances > 4 Å) needs to be revised. Our typical criteria for defining “contacts” are as follows:

van der Waals (hydrophobic) contacts: ≤ 4.5 Å

Hydrogen-bonds: ≤ 3.5 Å (combined with geometric considerations).

Ionic interactions (i.e. favorable interactions between oppositely-charged side chains, such as an Arg and Glu, or an Arg and the DNA phosphate backbone): ≤ 4.5 Å

We believe this is not grossly out of line with accepted practice.

In our text, “polar contacts” includes both H-bonds and ionic bonds. “Ionic bonds” can include both close range interactions (where the hydration shells of the two oppositely-charged moieties are displaced) that are typically called “salt-bridges” (these are typically ≤ 3.5 Å or so), but can also include longer range interactions where the two oppositely-charged moieties remain hydrated but their Coloumbic interaction is favorable [Xu, Tsai and Nussinov, 1997; Kumar and Nussinov, 2002; Yu, Pettitt and Iwahara, 2019]. These longer-range, favorable interactions are significant and extend to well beyond 4.5 Å distance [for instance, see Figure 1C of Yu, Pettitt and Iwahara, 2019]. Our choice of a 4.5 Å cutoff is actually quite conservative. We have clarified our notation of “polar interactions” (subsection “One ATP hydrolysis cycle corresponds to translocation by one base pair”, and in the Figure 3A legend).

We agree it was ambiguous how the center-of-gravity (cog) shift of TD2(ATP) to TD2(ADP) was calculated (i.e. exactly which structures were used). This is partly because the value for the shift (~3.5 Å) does not depend on how it is calculated (see below):

shift of TD2cog(ATP->ADP):

– using the average of all structures (L1/C1/C2/C5 -> L2/C3/C4): 3.56 Å

– using only confirmed structures (C1/C2/C5 -> C4): 3.54 Å

– using only highest resolution structures (C5 -> C4): 3.47 Å

We have clarified this point in the Figure 3B legend.

4) In the next step the TRG motif is compared to an example of residues R929, R953, and Q963. Here again, dashed lines in the figure exceeding a distance of 4 Å are misleading. A minor issue is the side chain orientation of Q963. The hydrogen bonding requirements are much better satisfied when the side chain is flipped; i.e., OE1 would interact with NZ of K739 instead of with the phosphate backbone and NE2 with the phosphate backbone. Although it is appreciated that there are a lot of residues to refine in total, groups that are used to exemplify certain findings should be modeled as accurately as possible.

Again, we do not agree that polar contacts exceeding 4 Å should be revised.

We thank the review for noticing the detail regarding the Q963 side chain – we have adjusted the position of Q963 accordingly in each PDB.

5) Regarding the statement: "As a result of the large conformational rearrangement of Mfd, Mfd(ADP)L2 is topologically “wrapped” around the DNA (Figure 5), likely explaining how Mfd (with RNAP in tow) translocates processively over many kilobases of DNA (Fan et al., 2016; Graves et al., 2015)." This claim is made without any further explanation or additional data supporting. Please clarify or remove.

We see a structural feature (Mfd topologically “wrapping” around the duplex DNA; Figure 5 and also see new Video 2) that goes a long way towards explaining previously observed (and well documented/established) biophysical (single-molecule) results (extremely high processivity; i.e. ability of one molecule of Mfd to translocate on many kb of duplex DNA without dissociating) of Mfd translocation subsequent to RNAP displacement (Fan et al., 2016; Graves et al., 2015).

The connection between topological wrapping or clamping of a nucleic acid processing enzyme on the nucleic acid template and high processivity is well accepted [see, for example, Breyer and Matthews, 2001]. Not all highly processive nucleic acid processing enzymes topologically wrap or clamp the nucleic acids, but wrapping or clamping the nucleic acids is often associated with what I would call “extremely” high processivity (thousands of nucleotides or even more). Examples include replicative DNA polymerases attached to their processivity clamps (PCNA in eukaryotes and archaea, Pol III b-subunit in bacteria, related structures in many bacteriophage) and cellular DNA-dependent RNA polymerases (which essentially have infinite processivity).

Thus, many DNA processing enzymes that topologically wrap or clamp onto the nucleic acid template have extremely high processivity. In its elaborate conformational transitions during the process of EC disruption, Mfd topologically wraps itself around the duplex DNA. Thus, it is natural to propose that the extremely high processivity of Mfd translocation is due to this topological wrapping.

We have explained this point more clearly in the Discussion.

6) If the discrimination between arrested and paused transcription complexes is solely kinetic, would it be true that an exceptionally long pause would enable Mfd dissociation? Please explain.

Transcriptional pausing in bacteria has been studied extensively in vitro, but these studies often utilize sub-saturating concentrations of NTPs to prolong pause lifetimes, making them easier to study. The distribution of pause lifetimes in vivo is not known (Robert Landick, personal communication), but the in vivo half-lives of some well-characterized long-lived pauses were estimated [Larson et al., 2014]: the His-pause half-life (known to be among the longest-lived pauses in vivo) was estimated to be ~1.3 sec.

Alternatively, single-molecule experiments of Howan et al., 2012, which were done under saturating ATP concentrations, determined that the mean lifetime of the Mfd-EC displacement complex (our L1 -> C5) between Mfd engagement (L1) and RNAP displacement (post-C5) is about 45 sec. This is a lower bound for the mean lifetime of a stalled EC due to the high Mfd concentration used in these experiments. Thus, the expected time required for Mfd to find and displace a stalled or paused EC is more than an order of magnitude longer than the longest-lived transcriptional pauses in vivo.

This issue is now addressed in the Discussion.

7) The subsection “Mfd loading requires multiple rounds of ATP hydrolysis and is accompanied by profound conformational changes” describes the translocation module TD1-TD2 "bumping" into RNA polymerase in the L2 structure. This presumably is different from the contact between the RID and the RNAP protrusion. Where does it make this contact, and is there a molecular definition of the bump? Is this related to the changes in the RNA polymerase clamp opening during the Mfd ATP cycle? Also, would the clamp opening cycle be different for paused versus blocked complexes, and could this contribute to the relative insensitivity of paused complexes to release by Mfd? Please clarify.

To clarify, we have added a new supplementary figure (Figure 5—figure supplement 1) that plots the interface areas between selected Mfd and RNAP domains for each of the Mfd-EC structures. From this plot we can see that the initial engagement of Mfd with the RNAP is through the Mfd-D4(RID):βprotrusion interaction (average interface area of 553 Å2) and that this does not change through all seven states (L1, L2, C1-C5).

In L1, the Mfd translocation module [D5(TD1)/D6(TD2)] interacts with upstream DNA (roughly -38 to -27) and does not interact with RNAP. Upon the transition to L2, the translocation module walks on the DNA, moving towards the RNAP until it “bumps” into the RNAP, characterized primarily by D6(TD2) also interacting with the RNAP βprotrusion [the Mfd-D4(RID) and D6(TD2) interfaces with the RNAP βprotrusion do not overlap]. Once Mfd-D6(TD2) pushes up against the RNAP βprotrusion, that interface is also maintained throughout the rest of the structures (L2, C1-C5, average interface area of 312 Å2).

This is now explained more clearly in the manuscript (subsection “Mfd loading requires multiple rounds of ATP hydrolysis and is accompanied by profound conformational changes”).

During the Mfd-NHC, the movements of Mfd relative to the RNAP cause cyclical RNAP conformational changes involving RNAP clamp and βlobe motions (Figure 7). The RNAP clamp is opened in C3 as Mfd wedges itself between the βprotrusion and the clamp, pushing on the clamp through a large interface with the translocation module (maximum Mfd-[D5(TD1)/D6(TD2)]:b'clamp interface area of 944 Å2 in C3). The RNAP βlobe is pushed sideways by an interaction with Mfd-D1 (can be seen in Figure 7). This is described in the subsection “Mfd manipulates the RNAP conformation during its nucleotide hydrolysis cycle”.

To our knowledge, the sensitivity of paused RNAP ECs to Mfd (other than backtrack pauses, which are sensitive to Mfd; Park et al., 2006) has not been specifically tested, but our argument above strongly supports the view that the relatively slow kinetics of Mfd engagement and displacement of ECs ensures that only stalled ECs that need to be rescued are actually displaced by Mfd.

8) Le et al., 2018, showed by single molecule analysis that Mfd can translocate by itself without engagement with RNA polymerase, and proposed a model in which Mfd patrols DNA, moving slower than transcription so that only stopped, or perhaps extensively paused, complexes can be captured. This does appear to be a distinct view of the cellular process. If the model of Le et al. describes the cellular process, some stochastic or other process presumably would have the break the inhibitory structure of the holo Mfd, and this would have to occur at a significant rate. Arguably, Le et al. could have been observing a small fraction of Mfd that is spontaneously active, and therefore not an important contributor to the cellular mechanism. Do the authors have any further thoughts about the two distinct models?

Although our pathway initiating through the hypothetical intermediate [L0] involves apo-Mfd (auto-inhibited) engaging directly with a stalled EC from solution and then either dissociating or transitioning into L1 to initiate the pathway, our results do not rule out a role for the “catch and release” model of Le et al., 2018. A statement to this effect has been added in the Discussion.

In our in vitro experiments, our EC scaffold only contains 50 base pairs of duplex DNA upstream of the transcription bubble, just enough for Mfd engagement (Park et al., 2006), suggesting our assays/structures all involve the direct pathway. A structure of Mfd actively translocating on duplex DNA on its own is not available, but we would presume that if a translocating Mfd were to encounter a stalled EC from the rear, it could engage with the EC and form a structure corresponding to our L1; the rest of the Mfd loading pathway would then proceed normally. The roles of the solution vs. translocation encounter pathways could be probed by the careful examination of EC-release kinetics on DNA scaffolds having increasing lengths of upstream duplex DNA. If Mfd molecules were able to engage with the duplex DNA and translocate on their own to catch up to and engage an EC, one could envision that increasing the length of the duplex DNA upstream of the stalled EC could increase the rate of EC release, but increasing the length of upstream duplex DNA beyond a certain length would no longer have an effect (due to the low processivity of the lone translocating Mfd).